# STOCHASTIC BANDITS ROBUST TO ADVERSARIAL ATTACKS

**Xuchuang Wang**
CICS, UMass Amherst

**Maoli Liu**
CSE, CUHK

**Jinhang Zuo**
CS, CityU

**Xutong Liu**
ECE, CMU

**John C.S. Lui**
CSE, CUHK

**Mohammad Hajiesmaili**
CICS, UMass Amherst

## ABSTRACT

This paper investigates stochastic bandit algorithms that are robust to adversarial attacks, where an attacker can first observe the learner's action and *then* alter their reward observation. We study two cases of this model, with or without the knowledge of an attack budget $C$, defined as an upper bound of the summation of the difference between the actual and altered rewards. For both cases, we devise two types of algorithms with regret bounds having additive or multiplicative $C$ dependence terms. For the known attack budget case, we prove our algorithms achieve the regret bound of $O((K/\Delta)\log T + KC)$ and $\tilde{O}(\sqrt{KTC})$ for the additive and multiplicative $C$ terms, respectively, where $K$ is the number of arms, $T$ is the time horizon, $\Delta$ is the gap between the expected rewards of the optimal arm and the second-best arm, and $\tilde{O}$ hides the logarithmic factors. For the unknown case, we prove our algorithms achieve the regret bound of $\tilde{O}(\sqrt{KT} + KC^2)$ and $\tilde{O}(KC\sqrt{T})$ for the additive and multiplicative $C$ terms, respectively. In addition to these upper bound results, we provide several lower bounds showing the tightness of our bounds and the optimality of our algorithms. These results delineate an intrinsic separation between the bandits with attacks and corruption models.

## 1 INTRODUCTION

Online learning literature (Borodin & El-Yaniv, 2005, §7.1.2) usually considers two types of non-oblivious adversary models: the medium adversary and the strong adversary.[1] The medium adversary chooses the next instance *before* observing the learner's actions, while the strong adversary chooses instances *after* observing the learner's actions. When it comes to the multi-armed bandits (MAB) learning with an adversary, the medium adversary corresponds to the bandits with corruption (Lykouris et al., 2018), and the strong adversary corresponds to adversarial attacks on bandits (Jun et al., 2018). Bandit algorithms robust to corruption are developed for the medium adversary models in bandits literature, e.g., Auer et al. (2002); Audibert & Bubeck (2010); Lykouris et al. (2018); Gupta et al. (2019). However, for the strong adversary model, i.e., the adversarial attack model on MAB, most previous efforts focus on devising attacking policies to mislead the learner to pull a suboptimal arm and thus suffer a linear regret, e.g., Jun et al. (2018); Liu & Shroff (2019); Zuo (2024). Developing robust algorithms for the adversarial attack model and achieving regret with benign dependence on the total attack is still largely unexplored (details discussed at the end of §1).

**Stochastic MAB with adversarial attacks** In this paper, we study the stochastic MAB under adversarial attacks and develop robust algorithms whose regret bounds degrade gracefully in the presence of such attacks. Denote by $K \in \mathbb{N}^+$ as the number of arms, and each arm $k$ is associated with a reward random variable $X_k$ with an unknown mean $\mu_k$. Denote by $k^*$ as the arm index with the highest mean reward, and $\Delta_k := \mu_{k^*} - \mu_k$ the reward gap between the optimal arm $k^*$ and any suboptimal arm $k$. The learner aims to minimize the *regret*, defined as the difference between the highest total rewards of pulling a single arm and the accumulated rewards of the concern algorithm.

---

[1]There is a third adversary model, called the weak adversary, which is oblivious. The medium adversary is also called the adaptive-online adversary, while the strong adversary is called the adaptive-offline adversary.

In each time slot, the learner first pulls an arm. Then, the adversary observes the pulled arm and its realized reward and chooses an attacked reward for the learner to observe. Denote by $C$ as the total amount of *attacks* that the adversary uses to alter the original rewards to the attacked rewards over all rounds. We present the extended formal model definitions in §2.

**Model difference between bandits with corruptions and attacks.** Different from the attack (strong adversary) above, in each time slot, the medium adversary (corruption) chooses the corrupted rewards for all arms *before* observing which arm is pulled by the learner. This simple order alternation yields intrinsic differences, making the attack model more challenging than the corruption model. Specifically, the attack model (attack after observing the pulled arm) makes the randomized arm pulling policies, widely used in algorithms for bandits robust to corruptions (Lykouris et al., 2018; Gupta et al., 2019), invalid. Without randomization, devising robust algorithms for attacks is much more challenging than that for corruptions. We discuss more details of the differences in §2.

**Contributions** This paper offers a thorough investigation into robust algorithms for stochastic MABs under an adversarial attack model. We explore both known and unknown attack budget $C$ scenarios, where the unknown attack budget case is also known as the adaptive budget case. For both cases, we develop robust algorithms with two kinds of regret bounds having additive or multiplicative $C$ dependence terms. We summarize our contributions as follows:

*Known attack budget case.* For the known attack case (§4), we first investigate the successive elimination with wider confidence radius (SE-WR) algorithm, which was initially proposed for bandits with corruptions (Lykouris et al., 2018). In §4.1, we show SE-WR can be applied to the bandits with known adversarial attack budget and prove that this algorithm enjoys a *tighter* regret upper bound $O(\sum_{k \neq k^*} \log T/\Delta_k + KC)$ (under both attack and corruption models) than previous analysis (Lykouris et al., 2018) under the corruption model. This improvement removes the original additive term $\sum_{k \neq k^*} C/\Delta_k$'s dependence of $\Delta_k$ and reduces it to $KC$. To achieve gap-independent regret bounds in §4.2, we propose two stopping conditions for the successive elimination to modify the SE-WR algorithm to two SE-WR-Stop algorithms (see ① in Figure 1). We prove that the regrets of these two modified algorithms are $O(\sqrt{KT \log T} + KC)$ and $O(\sqrt{KT(\log T + C)})$ respectively.

*Unknown attack budget case.* To address the MAB with unknown attack case (§5), we use algorithms proposed for known attacks as building blocks to devise algorithms for unknown attacks. We consider two types of algorithmic techniques. In §5.1, we apply a multi-phase idea to extend the SE-WR algorithm to a phase elimination algorithm (PE-WR, see ② in Figure 1). The PE-WR algorithm utilizes a multi-phase structure—each phase with carefully chosen attack budgets and phase lengths—to defend against unknown attacks. We show that the regret of PE-WR is upper bounded by $O(\sqrt{KT} + KC^2)$ (additive). In §5.2, we apply a model selection technique, which treats the unknown attack $C$ as a parameter in model selection (see ③ in Figure 1). Specifically, we consider $\log_2 T$ base instances of SE-WR-Stop with different input attack budgets and then use the model selection technique to "corral" these base instances, composing the model-selection SE-WR-Stop (called MS-SE-WR) algorithm. We upper bound the regret of MS-SE-WR by $\tilde{O}(\sqrt{KC}T^{\frac{2}{3}})$ or $\tilde{O}(KC\sqrt{T})$ (multiplicative) depending on the choice of the model selection algorithm. Figure 1 summarizes the relations of algorithm design, and Figure 2 shows which type of algorithm performs better, those with additive bounds or those with multiplicative bounds, varies depending on the total attacks, highlighting the necessity to study both types of bounds.

*Lower bounds.* In §3, we also provide lower bound results to show the tightness of our upper bounds. We first show a general $\Omega(KC)$ lower bound for any algorithm under attack. This lower bound improves a factor $K$ of the number of arms upon prior $\Omega(C)$ lower bounds (Gupta et al., 2019). Based on this, we propose the gap-dependent lower bound $\Omega(\sum_{k \neq k^*} \log T/\Delta_k + KC)$ (informal expression) and gap-independent lower bound $\Omega(\sqrt{KT} + KC)$. Our refined upper bound of SE-WR matches the gap-dependent lower bound up to some prefactors, and one of our proposed SE-WR-Stop algorithms matches the gap-independent lower bound up to some logarithmic factors. To show the tightness of our upper bounds for the unknown attack algorithms, we derive lower bounds, $\Omega(T^\alpha + C^{\frac{1}{\alpha}})$ and $\Omega(C^{\frac{1}{\alpha}-1}T^\alpha)$ with $\alpha \in [\frac{1}{2}, 1)$, for two classes of algorithms whose regret upper bounds follow certain additive and multiplicative forms respectively. With $\alpha = \frac{1}{2}$, these two lower bounds, becoming $\Omega(\sqrt{T} + C^2)$ and $\Omega(C\sqrt{T})$, match the upper bounds of PE-WR and MS-SE-WR in terms of $T$ and $C$. We summarize our analysis results in Table 1.

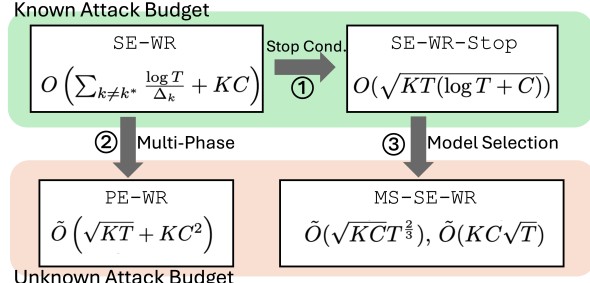

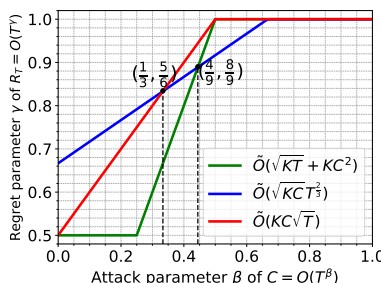

Figure 1: Algorithm design overview: only `SE-WR` has a gap-dependent bound; others are all gap-independent.

Figure 2: Comparison of unknown $C$ regrets (see Remark 11 for detail)

Table 1: Results overview: upper bounds with $^\dagger$ are for pseudo regret in expectation, while all other upper bounds are for realized regret with high probability; lower bounds with $^\ddagger$ are only for special classes of algorithms (see Propositions 4 and 5), and the lower bound with $^{\dagger\dagger}$ holds when $T \to \infty$. We study both additive and multiplicative bounds because which one is better depends on the value of $C$ (see Figure 2 for unknown $C$ case and the detailed discussion in Remark 11).

| Regret Bounds | Known Attack $C$ (§4) | Unknown Attack $C$ (§5) |
|---|---|---|
| **Additive** $C$ | $O\left(\sum_{k \neq k^*} \frac{\log T}{\Delta_k} + KC\right), O(\sqrt{KT \log T} + KC)$ | $\tilde{O}\left(\sqrt{KT} + KC^2\right)$ |
| **Multiplicative** $C$ | $O(\sqrt{KT(\log T + C)})$ | $\tilde{O}(\sqrt{KC}T^{\frac{2}{3}})^\dagger, \tilde{O}(KC\sqrt{T})^\dagger$ |
| **Lower Bounds** (§3) | $\Omega\left(\sum_{k \neq k^*} \frac{\log T}{\Delta_k} + KC\right)^{\dagger\dagger}, \Omega(\sqrt{KT} + KC)$ | $\Omega(T^\alpha + C^{\frac{1}{\alpha}})^\ddagger, \Omega(C^{\frac{1}{\alpha}-1}T^\alpha)^\ddagger$ |

*Experiment.* Besides the above analytical results, we also conduct experiments to validate the performance of our algorithms. The attacker employs the attack strategy proposed in Jun et al. (2018) when there are remaining attack budgets. We compare the regrets of our algorithms with prior algorithms under different attack budgets. The results show that our algorithms outperform the state-of-the-art algorithms in most cases, especially when the attack budget is large. We provide more details in §6.

**Result separation between corruption and attack.** With the results in this paper, we demonstrate a clear *separation* between corruption (medium adversary) and attack (strong adversary) models in terms of the additive regret bounds. Denote by $C'$ the total corruption to distinguish from the total attack $C$. In the case of known corruption/attack budget, while both achieving the gap-dependent regret bounds $\sum_{k \neq k^*}(1/\Delta_k) \log T$ in $T$-dependent terms, we show that the regret due to attack is $\Theta(KC)$ which has a factor of $K$ worse than the additional $\Theta(C')$ regret due to corruption. That implies, with the same amount of budget, an attacker can produce $K$ times larger impact on regret than that under the corruption model. For unknown budget cases, while bandits with corruptions have the regret bounds of $O(\sum_{k \neq k^*} \log^2(KT/\delta)/\Delta_k + KC')$ (Gupta et al., 2019), it is impossible for attack case to achieve a regret upper bound in the form of $O\left(\text{ploylog}(T) + C^\alpha\right)$ for any $\alpha > 0$ (contradicts with Theorem 3). Instead, we show that the regret with unknown attacks can attain $\tilde{O}(\sqrt{KT} + KC^2)$ where the $C^2$ additive term is tight. Even with ignoring the worse order of the first $T$-related terms, the second regret term due to attack $O(C^2)$ is still a factor of $C$ worse than the additional $O(C')$ regret due to corruption. That is, an attack with $O(\sqrt{T})$ budget is enough to make the algorithm suffer a linear regret in the attack model,[2] while one needs a linear $O(T)$ corruption budget to achieve the same effect in the corruption case. This separation is also studied in linear bandits in the concurrent work (Liu et al., 2024).

**Extended literature review** *Robustness against corruption.* Lykouris et al. (2018) propose the bandits with corruption model and devised algorithms with $O((K \log T + KC')/\Delta)$ regret for known corruption $C'$ case and algorithms with $O(C'K^2 \log T/\Delta)$ regret for unknown $C'$ case, where $\Delta$

---

[2]This statement is for the algorithm with $\tilde{O}(\sqrt{KT} + KC^2)$ mentioned above. Our lower bound results do not exclude the possible existence of other algorithms with upper bounds $\tilde{O}(T^\alpha + C^{\frac{1}{\alpha}})$ for $\alpha \in [\frac{1}{2}, 1)$, in which case a sublinear $O(T^\alpha)$ budget can make the algorithm suffer linear regret.

---

**Procedure 1** Decision under Attacks
1: **for** each round $t = 1, 2, \ldots, T$ **do**
2:     The learner pulls an arm $I_t$
3:     A stochastic reward $X_{I_t,t}$ is drawn from this arm $I_t$
4:     The adversary observes the pulled arm $I_t$ and its realized reward $X_{I_t,t}$
5:     The adversary chooses an attacked reward $\tilde{X}_{I_t,t}$
6:     The learner only observes the attacked reward $\tilde{X}_{I_t,t}$, and not $X_{I_t,t}$

---

denotes the smallest non-zero reward gap. Later, Gupta et al. (2019) study the same model and propose an algorithm that improves the multiplicative $C'$ dependence of regret for unknown $C'$ case to additive $C'$, i.e., $O(K \log^2 T/\Delta + KC')$. However, the robust design of both algorithms above relies on randomly pulling arms, which is not feasible in our attack setting. Besides bandits robust to corruption, there is another line of works on best-of-both-world bandit algorithm design, e.g., Seldin & Lugosi (2017); Wei & Luo (2018); Zimmert & Seldin (2021); Masoudian & Seldin (2021), where their algorithms can be applied to the bandit with corruption but do not work in the adversarial attack model either. Notably, Zimmert & Seldin (2021); Masoudian & Seldin (2021) show that their best-of-both-worlds algorithms can be applied to the bandit with corruption model and achieve $O(K \log T/\Delta + \sqrt{C' \log T})$ regret, which can be better than that of Gupta et al. (2019) in some parameter regimes.

*Attacks on bandits.* Attack policy design for MAB algorithms has been studied by Jun et al. (2018); Liu & Shroff (2019); Xu et al. (2021); Zuo et al. (2024); Zuo (2024) and many others. They aim to design attacking policies to mislead the learner, typically with logarithmic regrets when there is no attack, to pull a suboptimal arm and thus suffer a linear regret. This is the opposite of our objective for designing robust algorithms to achieve sublinear regret under any attacks.

*Robustness against attacks.* The only existing results on robust algorithms for MAB under attacks are by Zuo (2024, Section 6) and Rangi et al. (2022). Zuo et al. (2024, Section 6) uses competitive ratio as the objective—used when one cannot achieve sublinear regret, instead of the finer-grained regret studied in this paper. Rangi et al. (2022) studies how the "verification scheme", an additional assumption, can help the learner to detect the attack and achieve sublinear regret, but they do not study robust algorithms as this paper does. Apart from the MAB model, there are works studying robust algorithms on structured bandits under attacks, e.g., linear bandits (Bogunovic et al., 2021; He et al., 2022), Gaussian bandits (Bogunovic et al., 2020; 2022), and Lipschitz bandits (Kang et al., 2024). Among them, Bogunovic et al. (2020) are the first to devise robust algorithms for bandits under attacks (referred to as corruptions). Although their bandits results can be reduced to MAB, this reduction does not provide tight regrets as in Table 1 of this paper. We elaborate on this in §5.

## 2 MODEL: MAB WITH ADVERSARIAL ATTACKS

We consider a MAB with $K \in \mathbb{N}^+$ arms. Each arm $k \in \mathcal{K} := \{1, 2, \ldots, K\}$ is associated with a reward random variable $X_k \in [0, 1]$ with an unknown mean $\mu_k$. Denote the unique arm with highest reward mean as $k^*$, i.e., $k^* = \arg\max_{k \in \mathcal{K}} \mu_k$.[3] We consider $T \in \mathbb{N}^+$ decision rounds. At each round $t \in \mathcal{T} := \{1, 2, \ldots, T\}$, the learner selects an arm $I_t \in \mathcal{K}$, and the arm generates a reward realization $X_{I_t,t}$ (not disclosed to the learner yet). The adversary observes the pulled arm $I_t$ and its realized reward $X_{I_t,t}$ and then chooses an attacked reward $\tilde{X}_{I_t,t}$ for the learner to observe. The total attack is the sum of the absolute differences between the raw rewards and the attacked rewards over all rounds, i.e., $C := \sum_{t=1}^{T} |X_{I_t,t} - \tilde{X}_{I_t,t}|$. We summarize the decision process in Procedure 1.

**Regret objective** The adversary aims to maximize the learner's regret, while the learner aims to minimize the regret. We define the (realized) regret as the difference between the highest total realized *raw* rewards of a single arm and the accumulated *raw* rewards of the learning algorithm,

$$R_T := \max_{k \in \mathcal{K}} \sum_{t \in \mathcal{T}} (X_{k,t} - X_{I_t,t}). \tag{1}$$

---

[3]We assume the uniqueness of the optimal arm for simplicity. Our algorithms and results (with slightly change) also work for the case with multiple optimal arms.

Besides the realized regret in (1), another common regret definition is the pseudo-regret, defined as $\bar{R}_T := \max_{k \in \mathcal{K}} \mathbb{E}\left[\sum_{t \in \mathcal{T}}(X_{k,t} - X_{I_t,t})\right] = \mathbb{E}\left[T\mu_{k^*} - \sum_{t \in \mathcal{T}}\mu_{I_t}\right]$. By Jensen's inequality, we have $\bar{R}_T \leqslant \mathbb{E}[R_T]$, implying that the pseudo-regret is a weaker metric than the realized regret. The paper focuses on realized regret, with pseudo-regret results included as needed.

**Comparison between corruption and attack models** Unlike the attack model, prior work on robust bandit algorithms against corruption (Lykouris et al., 2018; Gupta et al., 2019) involves the adversary selecting corruption rewards *before* observing the learner's actions (see Appendix B for details). The total corruption is defined as $C' := \sum_{t=1}^{T} \max_k |X_t(k) - \tilde{X}_{k,t}|$. Although the definitions of corruption and attacks differ slightly, the models are fundamentally distinct due to the sequence of events: in the attack model, the adversary modifies rewards *after* observing the learner's actions, whereas in the corruption model, the modification occurs *before*.

To illustrate this distinction, consider that achieving the same impact on a bandit algorithm requires a significantly larger budget under the corruption model compared to the attack model. For example, a naive attack policy that forces the learner to incur linear regrets might involve altering the reward of the optimal arm $k^*$ whenever it is selected, making it smaller than the mean of the best suboptimal arm (Liu & Shroff, 2019). In the attack model, where the adversary observes the learner's action before attacking, this strategy may require only an $O(\log T)$ or sublinear $O(T^\alpha)$ budget for some $\alpha < 1$, depending on the specific algorithm. In contrast, under the corruption model, since the adversary does not know which arm will be pulled before corrupting the rewards, they would need to corrupt the optimal arm's reward realization in all $T$ rounds to achieve the same effect, resulting in an $O(T)$ cost. This is substantially higher than the sublinear attack budget. From another perspective, designing robust algorithms is more challenging in the attack model because the adversary's ability to act *after* observing the selected arm renders common randomized strategies ineffective. For instance, algorithms like EXP3.P (Auer et al., 2002) and BARBAR (Gupta et al., 2019) utilize randomization to mitigate corruption, but this approach is less effective against adversaries who can adapt their attacks based on observed actions. Therefore, developing robust algorithms against such adaptive adversaries is significantly more complex.

## 3 LOWER BOUNDS

This section first presents a general lower bound for the adversarial attack model on stochastic multi-armed bandits (Theorem 1) and two of its variants. Later, we derive two lower bounds for two special classes of bandit algorithms (Propositions 4 and 5). All proofs are deferred to Appendix C.

### 3.1 A GENERAL LOWER BOUND

We first present a general lower bound $\Omega(KC)$ in Theorem 1, originally derived in the linear bandit scenario by Bogunovic et al. (2021, Theorem 3). Together with known lower bounds of MAB, we derive Proposition 2. The §4 presents algorithms matching these lower bounds.

**Theorem 1.** *Given a stochastic multi-armed bandit game with $K$ arms, under attack with budget $C$, and $T > KC$ decision rounds,[4] for any bandits algorithm, there exists an attack policy with budget $C$ that can make the algorithm suffer $\Omega(KC)$ regret.*

**Proposition 2.** *For gap-dependent regret bounds and any consistent bandit algorithm[5], the lower bound is roughly $\Omega\left(\sum_{k \neq k^*} \frac{\log T}{\Delta_k} + KC\right)$, or formally, for some universal constant $\xi > 0$,*

$$\liminf_{T \to \infty} \frac{\bar{R}_T - \xi KC}{\log T} \geqslant \sum_{k \neq k^*} \frac{1}{2\Delta_k}. \tag{2}$$

*For gap-independent cases, the lower bound is*
$$\bar{R}_T \geqslant \Omega(\sqrt{KT} + KC). \tag{3}$$

### 3.2 TWO LOWER BOUNDS FOR SPECIAL ALGORITHM CLASSES

We first recall results from adversarial bandit attacks on bandits (Liu & Shroff, 2019; Zuo, 2024) in Theorem 3, then derive two lower bounds for bandit algorithms with additive and multiplicative regret bounds respectively. In §5, we present algorithms that match these lower bounds.

---

[4]Note that $T \leqslant KC$ implies that $C = \Omega(T)$ which trivially results in a linear regret.
[5]When $C = 0$, a consistent bandits algorithm has regret $\bar{R}_T \leqslant O(T^\alpha)$ for any $\alpha \in (0, 1)$.

---

**Algorithm 2** `SE-WR`: Successive Elimination with Wide Confidence Radius

---

**Input:** total attack $C$ (or budget $C_{\text{Input}}$), number of arms $K$, decision rounds $T$, parameter $\delta$
**Initialize:** candidate arm set $\mathcal{S} \leftarrow \mathcal{K}$, number of arm pulls $N_k \leftarrow 0$, reward estimates $\tilde{\mu}_k \leftarrow 0$
 1: **while** $t \leqslant T$ **do**
 2:      Uniformly pull each arm in $\mathcal{S}$ once and observes $\tilde{X}_k$ for all arms $k \in \mathcal{S}$
 3:      Update parameters: $t \leftarrow t + |\mathcal{S}|, N_k \leftarrow N_k + 1, \tilde{\mu}_k \leftarrow \frac{\tilde{\mu}_k(N_k-1)+\tilde{X}_k}{N_k}$ for all arms $k \in \mathcal{S}$
 4:      $\tilde{\mu}_{\max} \leftarrow \max_k \tilde{\mu}_k$
 5:      $\mathcal{S} \leftarrow \left\{ k \in \mathcal{S} : \tilde{\mu}_k \geqslant \tilde{\mu}_{\max} - 2\left( \sqrt{\frac{\log(2KT/\delta)}{N_k}} + \frac{C}{N_k} \right) \right\}$

---

**Theorem 3** (Adapted from (Zuo, 2024, Fact 1) and (He et al., 2022, Theorem 4.12)). *If an algorithm achieves regret $R_T$ when total attack $C = 0$, then there exists an attack policy with $C = \Theta(R_T)$ that can make the algorithm suffer linear regret.*

**Proposition 4** (Achievable additive regret bounds). *Given any parameter $\alpha \in [\frac{1}{2}, 1)$, for any bandit algorithm with an additive regret upper bound of the form $O(T^\alpha + C^\beta)$, we have $\beta \geqslant \frac{1}{\alpha}$.*

**Proposition 5** (Achievable multiplicative regret bounds). *Given parameter $\alpha \in [\frac{1}{2}, 1)$, for any bandit algorithm with a multiplicative regret bound of the form $O(T^\alpha C^\beta)$, we have $\beta \geqslant \frac{1}{\alpha} - 1$.*

## 4    ALGORITHMS WITH KNOWN ATTACK BUDGET

In this section, we study the stochastic multi-armed bandit problem with a known attack budget. In §4.1, we first present an algorithm with an additive gap-dependent upper bound on the attack budget. Then, in §4.2, we modify this algorithm to two variants with gap-independent upper bounds, one with an additive $C$ term and another with a multiplicative $C$ term.

### 4.1    SE-WR: AN ALGORITHM WITH GAP-DEPENDENT UPPER BOUND

Given the knowledge of attack budget $C$, one is able to predict the worst-case attack and design an algorithm to defend against it. Here, the robustness is achieved by widening the confidence radius of the reward estimate to account for the $C$ attack such that the corresponding widened confidence interval contains the true reward mean with a high probability (as if the rewards were not attacked). Denote $\tilde{\mu}_k$ as the empirical mean of the attacked reward observations of arm $k$. The widened confidence interval is centered at $\tilde{\mu}_k$ and has a radius of $\sqrt{\log(2/\delta)/N_k} + C/N_k$, where $N_k$ is the number of times arm $k$ is pulled and $\delta$ is the confidence parameter. Then, we utilize the successive elimination (`SE`) algorithm (Even-Dar et al., 2006) to devise a robust bandit algorithm in Algorithm 2. The algorithm maintains a candidate set of arms $\mathcal{S}$, initialized as the set of all arms $\mathcal{K}$, and successively eliminate suboptimal arms according to the widened confidence intervals (Line 5), called successive elimination with wide radius (`SE-WR`). A similar algorithm design idea was also used in Lykouris et al. (2018) for the stochastic bandit problem with corruption attacks. Theorem 6 provides the regret upper bound of Algorithm 2.

**Theorem 6.** *Given the total attack or its upper bound $C$, Algorithm 2's the realized regret is upper bounded by $R_T \leqslant O(\sum_{k \neq k^*} \log(KT/\delta)/\Delta_k + KC)$ with probability $1 - \delta$, and, with $\delta = K/T$, its pseudo regret is upper bounded as $\bar{R}_T \leqslant O(\sum_{k \neq k^*} \log T/\Delta_k + KC)$.*

The regret upper bound in Theorem 6 matches the lower bound in (2) in terms of both the first classic regret term and the second additive attack $\Omega(KC)$ term. This shows that the regret upper bound is nearly optimal up to some prefactors. Our regret upper bounds improve the $O(\sum_{k \neq k^*} \log(KT/\delta)/\Delta_k + \sum_{k \neq k^*} C/\Delta_k)$ in Lykouris et al. (2018, Theorem 1) by removing the multiplicative dependence of $1/\Delta_k$ on the total attack $C$ term. This improvement comes from a tighter analysis for sufficient pulling times for eliminating a suboptimal arm (see Lemma 14 in Appendix E). This improvement is crucial for achieving our tight multiplicative regret bounds for both known and unknown attack budget cases (Sections 4.2 and 5.2).

## 4.2 SE-WR-Stop: ALGORITHMS WITH GAP-INDEPENDENT UPPER BOUNDS

We follow the approach of stochastic bandits literature (Lattimore & Szepesvári, 2020, Chapter 6) to transfer the algorithm with a gap-dependent upper bound to another algorithm with gap-independent bounds. The main idea is to replace the while loop condition in Line 1 in Algorithm 2 with nuanced stopping conditions. If the stopping condition is triggered before the end of the MAB game, i.e., $t = T$, the algorithm will uniformly pull the remaining arms in the candidate set $\mathcal{S}$ until the end of the game. The pseudo-code is deferred to Algorithm 6 in Appendix D for its similarity to SE-WR.

We consider two kinds of stopping conditions. *The first condition* aims to transfer the gap-dependent bound to an independent one while keeping the additive $KC$ term in the upper bound. We start from a stopping condition $N_k \leqslant \log(KT/\delta)/\epsilon^2 + C/\epsilon$, where $\epsilon$ is a parameter to be determined. With this condition and the proof details of Theorem 6, we can upper bound the realized regret as follows, $R_T \leqslant O(K \log(KT/\delta)/\epsilon + \epsilon T + KC) \leqslant O(\sqrt{KT \log(KT/\delta)} + KC)$, where we choose $\epsilon = \sqrt{K \log(KT/\delta)/T}$ in the last inequality. *The second stopping condition,* while transferring the gap-dependent to independent, also converts the additive $KC$ term to multiplication. To derive it, we start from a stopping condition $N_k \leqslant \frac{\log(KT/\delta) + C}{\epsilon^2}$, where $\epsilon$ is a parameter to be determined. Similar to the derivation for the first condition, we can upper bound the realized regret as follows, $R_T \leqslant O((K \log(KT/\delta) + KC)/\epsilon + \epsilon T) \leqslant O(\sqrt{KT(\log(KT/\delta) + C)})$ where the last inequality sets $\epsilon = \sqrt{(K \log(KT/\delta) + KC)/T}$. The above results are summarized in Theorem 7 below.

**Theorem 7.** *Given the total attack or its upper bound $C$,* SE-WR-Stop *(Algorithm 6) with stopping condition $N_k \leqslant \frac{T}{K} + C\sqrt{\frac{T}{K \log(KT/\delta)}}$ has the realized regret upper bounded as follows,*

$$R_T \leqslant O(\sqrt{KT \log(KT/\delta)} + KC) \text{ with probability } 1 - \delta, \tag{4}$$

*and* SE-WR-Stop *with stopping condition $N_k \leqslant \frac{T}{K}$ has the realized regret as follows,*

$$R_T \leqslant O(\sqrt{KT(\log(KT/\delta) + C)}) \text{ with probability } 1 - \delta. \tag{5}$$

Let $\delta = K/T$, the regret in (4) becomes $O\left(\sqrt{KT \log T} + KC\right)$, matching the lower bound in (3) up to logarithmic factors. This implies the regret bound in (4) is nearly optimal. Let $\delta = K/T$, the regret in (5) becomes $O(\sqrt{KT(\log T + C)})$. Although this bound cannot match the lower bound in (3), its dependence on total attack $C$ is square root, better than the linear dependence in (4). This better dependence will shine when devising algorithms for unknown attack budgets in §5.2.

## 5 ALGORITHMS WITH UNKNOWN ATTACK BUDGET

This section presents algorithms that can defend against attacks with unknown attack budgets. We present algorithms with additive and multiplicative upper bounds in §5.1 and §5.2 respectively.

Before diving into the detailed algorithm designs, we recall that the input attack budget $C_{\text{Input}}$ for SE-WR and SE-WR-Stop can be interpreted as the level of robustness that the algorithms have. For any actual attack $C \leqslant C_{\text{Input}}$, the algorithms achieve the regret upper bounds in Theorems 6 and 7 respectively; for attack $C > C_{\text{Input}}$, the algorithms may be misled by the attacker and suffer linear regret. One simple way to achieve robustness against unknown attacks is to set the $C_{\text{Input}} = \sqrt{T/K}$ for SE-WR in §4.1, called SE-WR–$\sqrt{T/K}$ later, which leads to a piece-wise regret bound

$$R_T = \begin{cases} \tilde{O}(\sqrt{KT}) & \text{if } C = O(\sqrt{T/K}), \\ O(T) & \text{if } C = \Omega(\sqrt{T/K}). \end{cases} \tag{6}$$

We note that even this simple conversion-based bound in (6) is new among literature, thanks to our tighter analysis in Theorem 6. The best prior regret bound with a similar form is from He et al. (2022) (reduced to stochastic bandits): $R_T = \tilde{O}(K\sqrt{T})$ if $C = O(\sqrt{T})$; $R_T = O(T)$ if $C = \Omega(\sqrt{T})$, which is worse than our bound in (6) by a factor of $\sqrt{K}$. Without the knowledge of the actual attack budget $C$, the performance of SE-WR and SE-WR-Stop algorithms has a "robust-or-not" separation as in (6), which is not satisfactory for practical applications, especially when the actual corruption is small. In this section, we apply two types of "smoothing" algorithmic techniques to "smooth" this "robust-or-not" separation of SE-WR and SE-WR-Stop so as to achieve robustness against unknown attacks. An overview of algorithm design is in Figure 1 (see ② and ③).

---

**Algorithm 3** PE-WR: Phase Elimination with Wide Confidence Radius

---

**Input:** number of arms $K$, decision rounds $T$, confidence parameter $\delta$
**Initialize:** candidate arm set $\mathcal{S} \leftarrow \mathcal{K}$, the number of arm pulls $N_k \leftarrow 0$, reward mean estimates
  $\hat{\mu}_k \leftarrow 0$, phase $h \leftarrow 0$, phase upper bound $H \leftarrow \lceil \log_2 T \rceil - 1$, initial pulling times $L_0 \leftarrow K$
1: **for** each phase $h = 0, 1, \dots$ **do**
2:   $\hat{C}_h \leftarrow \min\{\sqrt{T}/K, 2^{H-h-1}\}$
3:   Pull each arm in $\mathcal{S}$ for $L_h/|\mathcal{S}_h|$ times
4:   Update arm empirical means $\hat{\mu}_{k,h}$ for all arms $k \in \mathcal{S}$
5:   $\mathcal{S}_{h+1} \leftarrow \left\{ k \in \mathcal{S}_h : \hat{\mu}_{k,h} \geqslant \max_{k' \in \mathcal{S}_h} \hat{\mu}_{k',h} - 2 \left( \sqrt{\frac{\log(2KH/\delta)}{L_h/|\mathcal{S}_h|}} + \frac{\hat{C}_h}{L_h/|\mathcal{S}_h|} \right) \right\}$
6:   $L_{h+1} \leftarrow 2L_h$

---

On the other hand, robust algorithms designed for unknown corruptions, e.g., Lykouris et al. (2018); Gupta et al. (2019), do not apply to the unknown attack setting. Because these robust algorithms all rely on some randomized action mechanism to defend against corruption, which is invalid for the attack as the attacker can manipulate the rewards of the arms *after* observing the learner's actions.

### 5.1 PE-WR: AN ALGORITHM WITH ADDITIVE UPPER BOUND

The first "smoothing" algorithmic technique is applying the multi-phase structure to modify SE-WR (Algorithm 2), yielding the phase elimination with wide confidence radius algorithm, PE-WR (Algorithm 3) for unknown attack budget. This algorithm considers multiple phases of SE-WR, each with a different assumed attack budget $C_{\text{Input}}$, and the candidate arm set $\mathcal{S}$ is updated consecutively among phases, that is, the arms eliminated in previous phases are not considered in later phases.

Specifically, PE-WR separates the total decision rounds into multiple phases $h \in \{1, 2, \dots\}$, each with a length doubled from the previous phase (Line 6). In phase $h$, the elimination assumes the potential attack is upper bounded by $\hat{C}_h$, which is halved in each phase (Line 2). This mechanism results in two kinds of phases. Denote $h'$ as the first phase whose corresponding $\hat{C}_{h'}$ is smaller than the true attack budget $C$. Then, for phases $h < h'$, we have $\hat{C}_h \geqslant C$, and, therefore, the algorithm eliminates arms properly as SE-WR; for phases $h \geqslant h'$, as $\hat{C}_h < C$, the algorithm may falsely eliminate the optimal arm and suffer extra regret. But even though the optimal arm is eliminated, the algorithm in these phases still has the level of $\hat{C}_h$ robustness, which guarantees some relatively good suboptimal arms—with a mean close to the optimal arm—remaining in the candidate set $\mathcal{S}$ such that the total regret due to losing the optimal arm in these phases is not large (bounded by $KC^2$ in the proof). We present PE-WR in Algorithm 3 and its regret upper bound in Theorem 8.

**Theorem 8.** *Algorithm 3 has a realized regret upper bound, with a probability of $1 - \delta$, $R_T \leqslant O(\sqrt{KT \log(K \log T/\delta)} + \sqrt{T} \log T + KC \log T + KC^2)$, and by setting $\delta = K/T$, a pseudo regret upper bound $\bar{R}_T \leqslant \tilde{O}(\sqrt{KT} + KC^2)$.*

Comparing our upper bound in Theorem 8 to the lower bound in Proposition 4 with $\alpha = 1/2$, $\Omega(\sqrt{T} + C^2)$, our upper bound is tight in terms of both $T$ and $C$. Compared with the lower bound $\Omega(\sqrt{KT} + KC)$ in Theorem 2, our upper bound is also tight in terms of $K$. We also note that the bound in Theorem 8 for PE-WR has the same order as the "robust-or-not" piece-wise bound in (6), but PE-WR is expected to be empirically superior, especially when the actual corruption is small.

This multi-phase elimination idea is widely used in bandit literature, e.g., batched bandits (Gao et al., 2019), multi-player bandits (Wang et al., 2019), and linear bandits (Bogunovic et al., 2021). The most related to ours is the phase elimination algorithm proposed in Bogunovic et al. (2021) for linear bandits. However, directly reducing their results (Bogunovic et al., 2021, Theorem 2) to our stochastic bandit case would yield a $\tilde{O}(\sqrt{KT} + KC\sqrt{K} \log T + C^2)$ upper bound for $C \leqslant \sqrt{T}/(K \log(\log K) \log T)$ only, and, for general $C$, their results would become $\tilde{O}(\sqrt{KT} + KC\sqrt{K} \log T + K^2 C^2)$. Our result in Theorem 8, simplified as $\tilde{O}(\sqrt{KT} + KC \log T + KC^2)$, is tighter than theirs by a factor of $\sqrt{K}$ on the second $\log T$ term, and, for the general $C$ case, our result is tighter by a factor of $K$ on the third $C^2$ term. This improvement comes from the tighter confidence bound employed in Line 5 of Algorithm 3 and the corresponding tighter analysis for the necessary number of observations for the algorithm to eliminate arms (Lemma 14) in §4.1.

---

**Algorithm 4** `MS-SE-WR`: Model Selection with Wide Confidence Radius

---

**Input:** number of arms $K$, decision rounds $T$, model selection algorithm `ModSelAlg`
  1: Construct $G := \lceil \log_2 T \rceil$ base algorithm instances as
$$\mathcal{B} \leftarrow \{\texttt{SE-WR-Stop}(C = 2^g, K, T, \delta = K/T) : g = 1, 2, \dots, G\}.$$
  2: `ModSelAlg`($\mathcal{B}$) for $T$ decision rounds

---

## 5.2 `MS-SE-WR`: An Algorithm with Multiplicative Upper Bound

While the additive bound in Theorem 8 can be transferred to a multiplicative bound $\tilde{O}(KC^2\sqrt{T})$, this bound cannot match the lower bound in Proposition 5, e.g., $\Omega(C\sqrt{T})$ for $\alpha = \frac{1}{2}$. In this section, we deploy another algorithmic technique, model selection, to `SE-WR-Stop` to achieve tight multiplicative upper bounds for the unknown attack budget case. Unlike the multi-phase technique, where each phase assumes a different attack budget, we consider multiple algorithm instances with different attack budget inputs, called "models." Model selection means selecting the model (algorithm instance) that performs best in the unknown environment. In our scenario, it is to select the instance with the smallest attack budget input $C_{\text{Input}}$ that is larger than the actual attack budget $C$.

Specially, we consider $G := \lceil \log_2 T \rceil$ instances of `SE-WR-Stop`, each with a different attack budget input $C_{\text{Input}} = 2^g$, $g = 1, 2, \dots, G$. Among these instances, the best is $g^* = \lceil \log_2 C \rceil$, as $2^{g^*} \in [C, 2C)$ is the smallest attack budget larger than the actual attack $C$. We then apply a model selection algorithm, e.g., `CORRAL` (Agarwal et al., 2017) or `EXP3.P` (Auer et al., 2002), to select the best one among the $G$ instances. Algorithm 4 presents the pseudo-code. To prove the regret upper bound, we recall a simplified version of Pacchiano et al. (2020, Theorem 5.3) in Lemma 9.

**Lemma 9.** *If base algorithm instance has regret upper bound $\bar{R}_T \leqslant T^\alpha c(\delta)$ with probability $1 - \delta$ for known constant $\alpha \in [\frac{1}{2}, 1)$ and the unknown function $c : \mathbb{R} \to \mathbb{R}$, then the regrets of `CORRAL` and `EXP3.P` are $\bar{R}_T \leqslant \tilde{O}(c(\delta)^{\frac{1}{\alpha}} T^\alpha)$ and $\bar{R}_T \leqslant \tilde{O}(c(\delta) T^{\frac{1}{2-\alpha}})$ respectively.*

Applying this lemma to our case, we first convert the gap-independent upper bound in (5) of `SE-WR-Stop` (Algorithm 6) in Theorem 7 to a multiplicative upper bound: When the stop condition is $N_k \leqslant \frac{T}{K}$, the realized regret upper bound is $\bar{R}_T \leqslant O(\sqrt{KTC \log(KT/\delta)})$ with a probability of at least $1 - \delta$. Plugging this into Lemma 9 with $\alpha = \frac{1}{2}$ and $c(\delta) = \sqrt{C \log(KT/\delta)}$ and letting $\delta = K/T$, we have the pseudo regret upper bounds for `MS-SE-WR` as follows:

**Theorem 10.** *Deploying `SE-WR-Stop` (stop condition $N_k \leqslant T/K$) as the base algorithm instance, Algorithm 4 with `CORRAL` has pseudo regret upper bound $\bar{R}_T \leqslant \tilde{O}(KC\sqrt{T})$, and Algorithm 4 with `EXP3.P` has pseudo regret upper bound $\bar{R}_T \leqslant \tilde{O}(\sqrt{KC}T^{\frac{2}{3}})$.*

The bounds in Theorem 10 are tight in terms of the trade-off of $T$ and $C$, as they respectively match the lower bounds in Proposition 5, $\Omega(\sqrt{C}T^{\frac{2}{3}})$ when $\alpha = \frac{2}{3}$ and $\Omega(C\sqrt{T})$ when $\alpha = \frac{1}{2}$. The challenge of achieving these tight regret bounds in Theorem 10 is in discovering the right base algorithm instances with suitable multiplicative upper bounds. For example, the first upper bound in Theorem 7 can also be converted to the multiplicative $\bar{R}_T \leqslant O(C\sqrt{KT \log(KT/\delta)})$ bound with a probability of at least $1 - \delta$. However, applying the model selection technique to this bound only results in $\tilde{O}(C\sqrt{K}T^{\frac{2}{3}})$ and $\tilde{O}(C^2 K\sqrt{T})$ bounds, which are not as tight as the ones in Theorem 10.

This model selection technique was also used in Kang et al. (2024) to boost the algorithm for Lipchitz bandits with a known attack budget to an algorithm with an unknown attack budget. However, their regret upper bounds, mapping to our `MAB` setting, are $\tilde{O}(C^{\frac{1}{K+2}} T^{\frac{K+2}{K+3}})$ and $\tilde{O}(C^{\frac{1}{K+1}} T^{\frac{K+1}{K+2}})$, which have a much worse dependence on $T$ than the ones in Theorem 10, especially when $K$ is large. This is because the regret upper bounds of their base algorithms have a much worse dependence on $T$ than our base `SE-WR-Stop` algorithm. This highlights the importance of devising the suitable base algorithm instances for `MAB` with the known attack budget in §4.2.

**Remark 11** (Regret comparison for unknown $C$ case in Figure 2). *The section presents three different upper bounds for unknown attacks, one additive $\tilde{O}(\sqrt{KT} + KC^2)$ by `PE-WR` and two multiplicative $\tilde{O}(\sqrt{KC}T^{\frac{2}{3}}), \tilde{O}(KC\sqrt{T})$ by two types of `MS-SE-WR`. Giving attack $C = T^\beta$ for some parameter $\beta \in [0, 1]$, these three bounds can be written in the form of $\tilde{O}(T^\gamma)$ for corresponding exponents $\gamma$. Figure 2 compares these bounds in terms of the exponent $\gamma$ by varying $\beta$. Comparing all three bounds, the additive bound is better than both multiplicative ones when $\beta \leqslant \frac{4}{9}$, and*

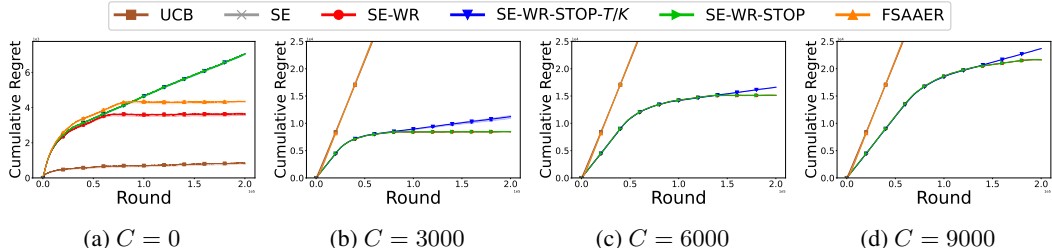

Figure 3: Regret comparison of algorithms with *known* attack budgets when varying budget $C$

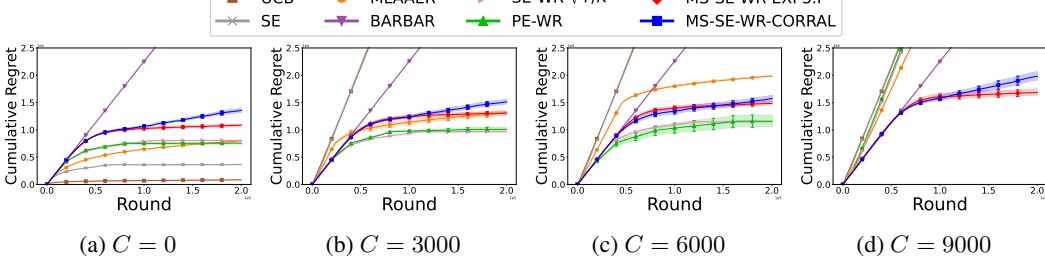

Figure 4: Regret comparison of algorithms with *unknown* attack budgets when varying budget $C$

*the* $\tilde{O}(\sqrt{KC}T^{\frac{2}{3}})$ *bound is better than the additive one when* $\beta > \frac{4}{9}$. Especially, the multiplicative bound $\tilde{O}(\sqrt{KC}T^{\frac{2}{3}})$ is still sublinear when the total corruption is large than $\sqrt{T}$, specifically, $\Omega(\sqrt{T}) < C < O(T^{\frac{2}{3}})$, which is not achievable by the additive bound. *Comparing among the two multiplicative bounds, the* $\tilde{O}(KC\sqrt{T})$ *is better than* $\tilde{O}(\sqrt{KC}T^{\frac{2}{3}})$ *when* $\beta \leqslant \frac{1}{3}$, *and vice versa.*

## 6 EXPERIMENTS

**Baselines** We compare our algorithms with baselines: Upper Confidence Bound (UCB) (Lattimore & Szepesvári, 2020), Successive Elimination (SE) (Even-Dar et al., 2006), Fast-Slow Active Arm Elimination Race (FSAAER) and Multi-Layer Active Arm Elimination Race (MLAAER) (Lykouris et al., 2018), BARBAR (Gupta et al., 2019). These baselines are grouped by whether they require knowledge of attack budgets. To distinguish our two SE-WR-Stop algorithms, we call the one with stopping condition $N_k \leqslant T/K$ as SE-WR-Stop–$T/K$ and the other SE-WR-Stop.

**Setup** We adopt the attack strategy from Jun et al. (2018) (details in Appendix G.1), a standard attack policy for stochastic bandits. Although our algorithms outperform baselines under this attack, they could perform even better under more sophisticated policies. We consider a stochastic bandit with $K = 10$ arms, where each arm $k$ follows a Bernoulli distribution with means $\mu_k$ decreasing from 0.9 to 0.45 in steps of 0.05. We evaluate all algorithms under four attack budgets ($C \in \{0, 3000, 6000, 9000\}$) over $T = 200000$ rounds, repeating each experiment 10 times and reporting the average and deviation of cumulative regret. Additional results can be found in Appendix G.

**Observations** Figures 3 and 4 show the regret of algorithms under known and unknown attack budgets. *Known attack budgets:* Figures 3b–3d show that baselines, including FSAAER (Lykouris et al., 2018), perform poorly, while our algorithms (SE-WR and SE-WR-Stop) achieve sublinear regret, confirming the need for developing robust algorithms in this attack model. The increase in the regret of SE-WR and SE-WR-Stop with the attack budget $C$ is consistent with our theoretical results. *Unknown attack budgets:* Figure 4 shows the following: (i) When total attack is small, e.g., $C = 0$ in Figure 4a, the adaptive PE-WR algorithm outperforms the SE-WR–$\sqrt{T/K}$ algorithm (with rigid "robust-or-not" separate in (6)), highlighting the need for adaptive strategies in unknown attack settings. (ii) Baselines like MLAAER (Lykouris et al., 2018) (developed for corruptions) achieve sublinear regret for small budgets (e.g., $C = 0$) as expected but perform poorly when $C = 6000$ (comparing to our robust algorithms, PE-WR and MS-SE-WR), and even fail when $C = 9000$, underscoring the need for devising robust algorithms for attacks. (iii) For large budgets ($C = 9000$), only our MS-SE-WR algorithms maintain sublinear regret, consistent with our theoretical results that algorithms with multiplicative regret bounds perform better when $C > \Omega(\sqrt{T})$ (detailed in Remark 11), emphasizing the importance of studying such bounds.

ACKNOWLEDGEMENTS

The authors want to thank Thodoris Lykouris for suggesting this topic in an AIM's SQuaREs meeting and for his invaluable feedback and insightful discussions throughout the research. The work of Jinhang Zuo was supported by CityU 9610706. The work of John C.S. Lui was supported in part by the RGC GRF14202923. The work of Mohammad Hajiesmaili was supported by NSF CPS-2136199, CAREER-2045641, and CNS-2325956. The work of Xutong Liu was supported in part by a fellowship award from the Research Grants Council of the Hong Kong Special Administrative Region, China (CUHK PDFS2324-4S04). Xutong Liu is the corresponding author.

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

## A  APPLICATION EXAMPLES OF OUR ATTACK MODEL

A key application of our model lies in online news recommendation systems. Here, the platform (learner) recommends articles (arms) to users with the goal of maximizing total clicks (rewards). However, click fraud botnets (adversaries) may maliciously target the system by generating fake clicks on specific articles (attack reward observations), thereby misleading the recommendation algorithm. This scenario extends naturally to other online recommendation systems, such as e-commerce, video streaming, and social media platforms.

Another potential application is in resource allocation for content delivery networks (CDNs). In this context, the CDN manager (learner) allocates resources (e.g., storage, bandwidth) to various servers (arms) to optimize overall performance (rewards, such as response time or throughput). However, the manager might encounter distributed denial-of-service (DDoS) attacks (adversaries) targeting specific servers (attack their reward observations) to degrade system performance.

## B  OTHER MODEL DETAILS: CORRUPTION PROCESS, REGRET DISCUSSION

In this section, we provide more details on the corruption process. At each round $t \in \{1, 2, \ldots, T\}$, the adversary observes the realized rewards $X_{k,t}$ for all arms $k$, as well as the rewards and actions of the learner in previous rounds. The adversary then chooses the corrupted rewards $\tilde{X}_{k,t}$ for all arms $k$. The learner pulls an arm $I_t$ (maybe randomly) and observes the attacked reward $\tilde{X}_{I_t,t}$. We summarize the decision process in Procedure 5. We denote the total amount of corruption as

$$C' := \sum_{t=1}^{T} \max_k |X_{k,t} - \tilde{X}_{k,t}|,$$

where $I_t$ is the pulled arm at time slot $t$.

---
**Procedure 5** Decision under Corruptions
---
1: **for** each round $t = 1, 2, \ldots, T$ **do**
2:     Stochastic rewards $X_{k,t}$ are drawn from all arms $k$
3:     The adversary observes the realized rewards $X_{k,t}$
                as well as the rewards and actions of the learner in previous rounds
4:     The adversary chooses the corruption rewards $\tilde{X}_{k,t}$ for all arms $k$
5:     The learner pulls an arm $I_t$ (maybe randomly) and observes the attacked reward $\tilde{X}_{I_t,t}$

---

**Remark 12** (Regret defined by corrupted rewards). *Beside the regret defined on raw reward in* (1)*, one can also define the regret based on the corrupted rewards as follows,* $\tilde{R}_T :=$ $\max_{k \in \mathcal{K}} \sum_{t \in \mathcal{T}} (\tilde{X}_{k,t} - \tilde{X}_{I_t,t})$*. This definition is considered in* Zimmert & Seldin (2021)*;* Bogunovic et al. (2020)*. As the adversary can manipulate at most $C$ rewards, one can easily show that,* $|\tilde{R}_T - R_T| \leqslant 2C$*. We will focus on the regret defined by raw rewards* (1) *in the following, as most results in this paper have at least a linear dependence on $C$.*

## C  PROOF FOR LOWER BOUND

**Theorem 1.** *Given a stochastic multi-armed bandit game with $K$ arms, under attack with budget $C$, and $T > KC$ decision rounds,*[6] *for any bandits algorithm, there exists an attack policy with budget $C$ that can make the algorithm suffer $\Omega(KC)$ regret.*

*Proof of Theorem 1.* We consider $K$ instances of a MAB game where for instance $\mathcal{I}_k$ for any $k \in \{1, 2, \ldots, K\}$, the arm $k$ has reward $\epsilon > 0$ and all other arms have reward $0$ without any noise.

The adversary's policy is that whenever the algorithm pulls an arm with a non-zero reward, the adversary attacks the algorithm by changing the reward of the arm to $0$. The adversary can conceal the optimal arm for $\lfloor \frac{C}{\epsilon} \rfloor$ pulls.

---

[6]Note that $T \leqslant KC$ implies that $C = \Omega(T)$ which trivially results in a linear regret.

After $\lfloor \frac{C}{\epsilon} \rfloor \lfloor \frac{K}{2} \rfloor$ time slots, there exists at least $\lfloor \frac{K}{2} \rfloor$ arms that are pulled at most $\lfloor \frac{C}{\epsilon} \rfloor$ times. That is, if the optimal arm is among these $\lfloor \frac{K}{2} \rfloor$ arms, the algorithm cannot identify the optimal arm. Let us pick the instance whose optimal arm is among these $\lfloor \frac{K}{2} \rfloor$ arms. Then, during these $\lfloor \frac{C}{\epsilon} \rfloor \lfloor \frac{K}{2} \rfloor$ time slots, the algorithm needs to pay the regret of

$$\left\lfloor \frac{C}{\epsilon} \right\rfloor \left( \left\lfloor \frac{K}{2} \right\rfloor - 1 \right) \epsilon = \Omega(KC).$$

$\square$

**Proposition 2** *For gap-dependent regret bounds and any consistent bandit algorithm[7], the lower bound is roughly* $\Omega\left( \sum_{k \neq k^*} \frac{\log T}{\Delta_k} + KC \right)$, *or formally, for some universal constant $\xi > 0$,*

$$\liminf_{T \to \infty} \frac{\bar{R}_T - \xi KC}{\log T} \geqslant \sum_{k \neq k^*} \frac{1}{2\Delta_k}.$$

*For gap-independent cases, the lower bound is*

$$\bar{R}_T \geqslant \Omega(\sqrt{KT} + KC).$$

*Proof of Proposition 2.* From bandits literature, e.g., Lattimore & Szepesvári (2020, Chapter 16), we know that the pseudo regret of any consistent bandit algorithm can be lower bounded as follows,

$$\liminf_{T \to \infty} \frac{\bar{R}_T}{\log T} \geqslant \sum_{k \neq k^*} \frac{1}{\Delta_k}.$$

On the other hand, we can rewrite the $\Omega(KC)$ lower bound in Theorem 1 as follows,

$$\liminf_{T \to \infty} \frac{\bar{R}_T - \xi KC}{\log T} \geqslant 0,$$

where $\xi > 0$ is a universal constant. Adding the above two inequalities and dividing both sides by 2, we have

$$\liminf_{T \to \infty} \frac{\bar{R}_T - (\xi/2)KC}{\log T} \geqslant \sum_{k \neq k^*} \frac{1}{2\Delta_k}.$$

For the gap-independent case, recall in bandits literature (Lattimore & Szepesvári, 2020, Chapter 15), we have the $\bar{R}_T \geqslant \Omega(\sqrt{KT})$ lower bound. Combining it with the $\Omega(KC)$ lower bound in Theorem 1 yields $\bar{R}_T \geqslant \Omega(\max\{\sqrt{KT}, KC\}) = \Omega(\sqrt{KT} + KC)$ lower bound. $\square$

**Proposition 4** (Achievable additive regret bounds) *Given any parameter $\alpha \in [\frac{1}{2}, 1)$, for any bandit algorithm that achieves an additive regret upper bound of the form $O(T^\alpha + C^\beta)$, we have $\beta \geqslant \frac{1}{\alpha}$.*

**Proposition 5** (Achievable multiplicative regret bounds) *Given parameter $\alpha \in [\frac{1}{2}, 1)$, for any bandit algorithm that achieves a multiplicative regret bound of the form $O(T^\alpha C^\beta)$, we have $\beta \geqslant \frac{1}{\alpha} - 1$.*

*Proof of Propositions 4 and 5.* We use contradiction to prove both propositions. We first prove Proposition 4. Given Theorem 3, we know that there exists an attack policy with $C = \Theta(R_T)$ that can make the algorithm suffer linear regret. Thus, if the algorithm achieves an additive regret upper bound of the form $O(T^\alpha + C^\beta)$ with $\beta < \frac{1}{\alpha}$, then the algorithm only suffers a sublinear regret when $C = T^\alpha$, which contradicts Theorem 3. Thus, the parameter $\beta \geqslant \frac{1}{\alpha}$. Proposition 5 can be proved via a similar contradiction argument. $\square$

**Algorithm 6** SE-WR-Stop: Successive Elimination with Wide Confidence Radius and Stop Condition

**Input:** total attack $C$, number of arms $K$, decision rounds $T$, confidence parameter $\delta$

1: Initialize candidate arm set $\mathcal{S} \leftarrow \mathcal{K}$, the number of arm pulls $N_k \leftarrow 0$, and reward mean estimates $\hat{\mu}_k \leftarrow 0$
2: **while** $N_k \leqslant \frac{T}{K}$ and $t \leqslant T$ **do**  $\qquad\qquad\qquad\qquad \triangleright$ or $N_k \leqslant \frac{T}{K} + C\sqrt{\frac{T}{K \log(KT/\delta)}}$
3: $\qquad$ Uniformly pull each arm in $\mathcal{S}$ once and observes $X_k$ for all arms $k \in \mathcal{S}$
4: $\qquad$ Update parameters: $t \leftarrow t + |\mathcal{S}|, N_k \leftarrow N_k + 1, \hat{\mu}_k \leftarrow \frac{\hat{\mu}_k(N_k-1)+X_k}{N_k}$
5: $\qquad$ $\hat{\mu}_{\max} \leftarrow \max_k \hat{\mu}_k$
6: $\qquad$ $\mathcal{S} \leftarrow \left\{ k \in \mathcal{S} : \hat{\mu}_k \geqslant \hat{\mu}_{\max} - 2\left( \sqrt{\frac{\log(2KT/\delta)}{N_k}} + \frac{C}{N_k} \right) \right\}$
7: Uniformly pull arms in $\mathcal{S}$ till the end of the game

## D  Deferred Algorithm Pseudo-Code

## E  Proof for Upper Bounds with Known Attack

**Theorem 6.** *Given the total attack or its upper bound $C$, Algorithm 2's the realized regret is upper bounded by $R_T \leqslant O(\sum_{k \neq k^*} \log(KT/\delta)/\Delta_k + KC)$ with probability $1 - \delta$, and, with $\delta = K/T$, its pseudo regret is upper bounded as $\bar{R}_T \leqslant O(\sum_{k \neq k^*} \log T/\Delta_k + KC)$.*

*Proof of Theorem 6.* We first recall the following lemma from Lykouris et al. (2018).

**Lemma 13** (Lykouris et al. (2018, Lemma 3.1)). *With a probability of at least $1 - \delta$, the optimal arm $k^*$ is never eliminated.*

Next, we prove a lemma shows the sufficient number of pulling that are necessary for a suboptimal arm to be eliminated. This lemma is tighter than the one in Lykouris et al. (2018, Lemma 3.2) (proved for $N_k \geqslant \frac{64 \log(2KT/\delta)+64C}{\Delta_k^2}$), and, based on it, we derive tighter regret bounds.

**Lemma 14.** *With a probability of at least $1 - \delta$, all suboptimal arms $k \neq k^*$ are eliminated after pulling each arm $N_k \geqslant \frac{64 \log(2KT/\delta)}{\Delta_k^2} + \frac{6C}{\Delta_k}$ times.*

*Proof for Lemma 14.* With Lemma 13, we know that the optimal arm $k^*$ is never eliminated. Thus, in the following, we show that the suboptimal arm $k$ will be eliminated by the optimal arm $k^*$ on or before $N_k \geqslant \frac{64 \log(2KT/\delta)}{\Delta_k^2} + \frac{8C}{\Delta_k}$.

Denote $\hat{\mu}_k$ as the empirical mean of the raw stochastic reward observation of arm $k$. By Hoeffding's inequality, we have, with a probability of at least $1 - \frac{\delta}{KT}$,

$$|\hat{\mu}_k - \mu_k| \leqslant \sqrt{\frac{\log(2KT/\delta)}{N_k}} \tag{7}$$

$$\leqslant \frac{\Delta_k}{8},$$

where the last inequality is due to $N_k \geqslant \frac{64 \log(2KT/\delta)}{\Delta_k^2}$.

Comparing the true empirical mean $\hat{\mu}_k$ with the attacked empirical mean $\mu_k$, we have

$$|\tilde{\mu}_k - \hat{\mu}_k| \leqslant \frac{C}{N_k} \leqslant \frac{\Delta_k}{8},$$

---
[7]When $C = 0$, a consistent bandits algorithm has regret $\bar{R}_T \leqslant O(T^\alpha)$ for any $\alpha \in (0, 1)$.

where the last inequality is due to $N_k \geqslant \frac{8C}{\Delta_k}$. Combining the two inequalities, we have

$$|\tilde{\mu}_k - \mu_k| \leqslant \frac{\Delta_k}{4}.$$

Last, we show that the elimination condition for arm $k$ would have been triggered before $N_k \geqslant \frac{64 \log(2KT/\delta)}{\Delta_k^2} + \frac{8C}{\Delta_k}$ as follows,

$$\tilde{\mu}_{k^*} - 2\left(\sqrt{\frac{\log(2KT/\delta)}{N_k}} + \frac{C}{N_k}\right) \geqslant \tilde{\mu}_{k^*} - 2\left(\frac{\Delta_k}{8} + \frac{\Delta_k}{8}\right)$$

$$\geqslant \mu_{k^*} - \frac{\Delta_k}{4} - \frac{2\Delta_k}{4}$$

$$= \mu_k + \frac{\Delta_k}{4}$$

$$\geqslant \tilde{\mu}_k.$$

$\square$

As the pseudo regret is upper bounded at most $\sum_k \Delta_k N_k$, we have

$$\bar{R}_T \leqslant \sum_{k \neq k^*} \Delta_k N_k$$

$$\leqslant \sum_{k \neq k^*} \Delta_k \left(\frac{64 \log(2KT/\delta)}{\Delta_k^2} + \frac{8C}{\Delta_k}\right)$$

$$= O\left(\sum_{k \neq k^*} \frac{\log(KT/\delta)}{\Delta_k} + KC\right)$$

$$\leqslant O\left(\sum_{k \neq k^*} \frac{\log T}{\Delta_k} + KC\right), \tag{8}$$

where the last inequality is by choosing $\delta = \frac{K}{T}$.

**Transfer pseudo-regret to realized regret.** Next, we show the high probability bound for the realized regret $R_T$. Multiplying $N_k$ to both sides of (7), we have that, for any arm $k$, the difference between the actual accumulated reward and its expected counterpart is upper bounded by $\sqrt{N_k \log(2KT/\delta)}$ with a probability of at least $1 - \delta/KT$. The probability of any of the above events does not hold is at most $1 - \delta$.

For any suboptimal arm $k$, we bound the difference as follows,

$$\sqrt{N_k \log(2KT/\delta)} \leqslant N_k \sqrt{\frac{\log(2KT/\delta)}{N_k}} \leqslant N_k \Delta_k \sqrt{\frac{\log(2KT/\delta)}{64 \log(2KT/\delta) + 8C\Delta_k}} \leqslant \frac{N_k \Delta_k}{8}.$$

In case that the arm $k'$ with the highest empirical mean is not the optimal arm $k^*$, for example, this arm $k'$ has a smaller reward gap $\Delta_k \leqslant O(\sqrt{1/T})$, the regret reduction due to this event is at most $N_{k'}\Delta_{k'}$.

Putting the potential impact of different kinds of reward realization above, we can bound the realized regret by $O(\sum_{k \neq k^*} N_k \Delta_k)$ still. Thus, with similar derivation as (8), we have the realized regret of the algorithm is at most

$$O\left(\sum_{k \neq k^*} \frac{\log(KT/\delta)}{\Delta_k} + KC\right) \text{ with probability } 1 - \delta.$$

$\square$

# F  PROOF FOR UPPER BOUNDS WITH UNKNOWN ATTACK

**Theorem 8.** *Algorithm 3 has a realized regret upper bound, with a probability of $1 - \delta$, $R_T \leqslant O(\sqrt{KT \log (K \log T/\delta)} + \sqrt{T} \log T + KC \log T + KC^2)$, and by setting $\delta = K/T$, a pseudo regret upper bound $\bar{R}_T \leqslant \tilde{O}(\sqrt{KT} + KC^2)$.*

*Proof of Theorem 8.* For $C > \sqrt{T}/K$, the $\tilde{O}(\sqrt{KT} + KC^2) = O(T)$ upper bound holds trivially. Hence, we only consider $C \leqslant \sqrt{T}/K$ in this proof.

Note that the elimination condition in Algorithm 3 fails with a probability of at most $\delta/KH$. Applying a union bound to sum all potential failure probabilities shows that, with a probability of at least $1 - \delta$, the elimination condition works properly for all arms in all phases. In the rest of the proof, we exclude this failure probability and assume that the elimination only fails when $C > \hat{C}_h$.

Denote $H'$ as the actual number of phases, while recall $H = \lceil \log_2 T \rceil - 1$ is the upper bound of the number of phases. As the analysis of Algorithm 2 shows, for phase $h$ with $\hat{C}_h > C$, the optimal arm $k^*$ is in the candidate set $\mathcal{S}_h$ with high probability. Denote $h'$ as the first phase with $\hat{C}_{h'} \leqslant C$, implying $H - h' \leqslant \log_2 C$.

Denote $k_h^*$ as the arm with the highest reward mean in phase $h$. Then, we have $k_h^* = k^*$ for $h < h'$. Next, we use induction to show, for $h \geqslant h'$, the remaining best arm $k_h^*$ is close to the optimal arm $k^*$ as follows,

$$\mu_{k^*} - \mu_{k_h^*} \leqslant \frac{2C}{L_h/|\mathcal{S}_h|}.$$

If the optimal arm is eliminated in phase $h'$, then we have

$$\hat{\mu}_{k^*,h'} \leqslant \max_{k \in \mathcal{S}_{h'}} \hat{\mu}_{k,h'} - 2 \left( \sqrt{\frac{\log(2KH/\delta)}{L_{h'}/|\mathcal{S}_{h'}|}} + \frac{\hat{C}_{h'}}{L_{h'}/|\mathcal{S}_{h'}|} \right).$$

Denote $\hat{k}_{h'}^* = \arg\max_{k \in \mathcal{S}_{h'}} \hat{\mu}_{k,h'}$. We have

$$
\begin{aligned}
\mu_{\hat{k}_{h'}^*} &\overset{(a)}{\geqslant} \hat{\mu}_{\hat{k}_{h'}^*,h'} - \left( \sqrt{\frac{\log(2KH/\delta)}{L_{h'}/|\mathcal{S}_{h'}|}} + \frac{C}{L_{h'}/|\mathcal{S}_{h'}|} \right) \\
&\overset{(b)}{\geqslant} \hat{\mu}_{k^*,h'} + 2 \left( \sqrt{\frac{\log(2KH/\delta)}{L_{h'}/|\mathcal{S}_{h'}|}} + \frac{\hat{C}_{h'}}{L_{h'}/|\mathcal{S}_{h'}|} \right) - \left( \sqrt{\frac{\log(2KH/\delta)}{L_{h'}/|\mathcal{S}_{h'}|}} + \frac{C}{L_{h'}/|\mathcal{S}_{h'}|} \right) \\
&\overset{(c)}{\geqslant} \mu_{k^*} + 2 \left( \sqrt{\frac{\log(2KH/\delta)}{L_{h'}/|\mathcal{S}_{h'}|}} + \frac{\hat{C}_{h'}}{L_{h'}/|\mathcal{S}_{h'}|} \right) - 2 \left( \sqrt{\frac{\log(2KH/\delta)}{L_{h'}/|\mathcal{S}_{h'}|}} + \frac{C}{L_{h'}/|\mathcal{S}_{h'}|} \right) \\
&= \mu_{k^*} - \frac{2(C - \hat{C}_{h'})}{L_{h'}/|\mathcal{S}_{h'}|} \\
&\geqslant \mu_{k^*} - \frac{2C}{L_{h'}/|\mathcal{S}_{h'}|} \\
&\geqslant \mu_{k^*} - \frac{2CK}{L_{h'}}
\end{aligned}
$$

where inequalities (a) and (c) come from the Hoeffding's inequality and the attack $C$, inequality (b) is due to the elimination condition in round $h'$. That is,

$$\mu_{k^*} - \mu_{\hat{k}_{h'}^*} \leqslant \frac{2CK}{L_{h'}}.$$

With a similar analysis, we can show, $\mu_{\hat{k}_{h'+i}^*} - \mu_{\hat{k}_{h'+i+1}^*} \leqslant \frac{2CK}{L_{h'+i}}$, for all $i = 1, \ldots, H - h' - 1$. Based on the inequality, we further have, for any $h \geqslant h'$,

$$\mu_{k^*} - \mu_{\hat{k}_h^*} \leqslant \sum_{i=0}^{h-h'} \frac{2CK}{L_{h'+i}} = \sum_{i=0}^{h-h'} \frac{2CK}{L_0 2^{h'+i}} \leqslant \frac{2CK}{L_0 2^{h'+1}}. \tag{9}$$

Next, we decompose the pseudo regret as follows,

$$\bar{R}_T = \sum_{t=1}^{T}(\mu_{k^*} - \mu_{I_t}) = \sum_{k \in \mathcal{K}} N_{k,T}(\mu_{k^*} - \mu_k) = \sum_{h=0}^{H'} \sum_{k \in \mathcal{S}_h} \frac{L_h}{|\mathcal{S}_h|}(\mu_{k^*} - \mu_k)$$

$$\leqslant L_0 + \sum_{h=1}^{h'} \sum_{k \in \mathcal{S}_h} \frac{L_h}{|\mathcal{S}_h|}(\mu_{k^*} - \mu_k) + \sum_{h=h'+1}^{H'} \sum_{k \in \mathcal{S}_h} \frac{L_h}{|\mathcal{S}_h|}(\mu_{k^*} - \mu_k)$$

$$= L_0 + \underbrace{\sum_{h=1}^{h'} \sum_{k \in \mathcal{S}_h} \frac{L_h}{|\mathcal{S}_h|}(\mu_{k^*} - \mu_k) + \sum_{h=h'+1}^{H'} \sum_{k \in \mathcal{S}_h} \frac{L_h}{|\mathcal{S}_h|}(\mu_{k_h^*} - \mu_k)}_{\text{Part I}}$$

$$+ \underbrace{\sum_{h=h'+1}^{H'} \sum_{k \in \mathcal{S}_h} \frac{L_h}{|\mathcal{S}_h|}(\mu_{k^*} - \mu_{k_h^*})}_{\text{Part II}}.$$

To bound Part I, we notice that all arms in $\mathcal{S}_h$ are not eliminated at the end of the previous phase, implying, for $h < h'$,

$$\mu_k \overset{(a)}{\geqslant} \hat{\mu}_k - \left(\sqrt{\frac{\log(2KH/\delta)}{L_{h-1}/|\mathcal{S}_{h-1}|}} + \frac{C}{L_{h-1}/|\mathcal{S}_{h-1}|}\right)$$

$$\overset{(b)}{\geqslant} \max_{k' \in \mathcal{S}_{h-1}} \hat{\mu}_{k',h-1} - 2\left(\sqrt{\frac{\log(2KH/\delta)}{L_{h-1}/|\mathcal{S}_{h-1}|}} + \frac{\hat{C}_{h-1}}{L_{h-1}/|\mathcal{S}_{h-1}|}\right)$$

$$- \left(\sqrt{\frac{\log(2KH/\delta)}{L_{h-1}/|\mathcal{S}_{h-1}|}} + \frac{C}{L_{h-1}/|\mathcal{S}_{h-1}|}\right)$$

$$\overset{(c)}{\geqslant} \hat{\mu}_{k^*,h-1} - \left(3\sqrt{\frac{\log(2KH/\delta)}{L_{h-1}/|\mathcal{S}_{h-1}|}} + \frac{2\hat{C}_{h-1} + C}{L_{h-1}/|\mathcal{S}_{h-1}|}\right)$$

$$\overset{(d)}{\geqslant} \mu_{k^*} - \left(4\sqrt{\frac{\log(2KH/\delta)}{L_{h-1}/|\mathcal{S}_{h-1}|}} + \frac{2\hat{C}_{h-1} + 2C}{L_{h-1}/|\mathcal{S}_{h-1}|}\right),$$

where inequalities (a) and (d) are due to the Hoeffding's inequality and the attack $C$, inequality (b) is due to the elimination condition, and inequality (c) is due to $k^* \in \mathcal{S}_{h-1}$. That is, for arms $k \in \mathcal{S}_h$, we have

$$\mu_{k^*} - \mu_k \leqslant 4\sqrt{\frac{\log(2KH/\delta)}{L_{h-1}/|\mathcal{S}_{h-1}|}} + \frac{2(\hat{C}_{h-1} + C)}{L_{h-1}/|\mathcal{S}_{h-1}|}. \tag{10}$$

As we have $k_h^* \in \mathcal{S}_h$, and with the same derivation as (10), we also have

$$\mu_{k_h^*} - \mu_k \leqslant 4\sqrt{\frac{\log(2KH/\delta)}{L_{h-1}/|\mathcal{S}_{h-1}|}} + \frac{2(\hat{C}_{h-1} + C)}{L_{h-1}/|\mathcal{S}_{h-1}|}. \tag{11}$$

Therefore, we have

$$
\sum_{h=1}^{h'}\sum_{k\in\mathcal{S}_h}\frac{L_h}{|\mathcal{S}_h|}\Delta_k + \sum_{h=h'+1}^{H'}\sum_{k\in\mathcal{S}_h}\frac{L_h}{|\mathcal{S}_h|}(\mu_{k_h^*}-\mu_k)
$$

$$
\overset{(a)}{\leqslant}\sum_{h=1}^{H'}L_h\cdot 4\sqrt{\frac{\log(2KH/\delta)}{L_{h-1}/|\mathcal{S}_{h-1}|}} + \frac{2(\hat{C}_{h-1}+C)}{L_{h-1}/|\mathcal{S}_{h-1}|}.
$$

$$
=\sum_{h=1}^{H'}\left(4\sqrt{\frac{L_h^2\log(2KH/\delta)}{L_{h-1}/|\mathcal{S}_{h-1}|}} + \frac{2L_h(\hat{C}_{h-1}+C)}{L_{h-1}/|\mathcal{S}_{h-1}|}\right) \tag{12}
$$

$$
\leqslant\sum_{h=1}^{H'}\left(4\sqrt{2L_h K\log(2KH/\delta)} + 4K(\hat{C}_{h-1}+C)\right)
$$

$$
=4\sqrt{2K\log(2KH/\delta)}\sum_{h=1}^{H'}\sqrt{L_h} + 4K\sum_{h=1}^{H'}(\hat{C}_{h-1}+C)
$$

$$
\overset{(a)}{\leqslant}4g\sqrt{2KL_0\log(2KH/\delta)T} + 4\sqrt{T}\log_2 T + 4KC\log_2 T.
$$

where inequality (a) is due to (10) and (11), and inequality (b) is because $\sum_{h=1}^{H'}\sqrt{L_h}\leqslant g\sqrt{T}$ since the summation of a geometric increasing series is upper bounded by $g\sqrt{L_{H'}}$, for some universal constant $g>0$, and $L_{H'}\leqslant T$.

Next, we bound Part II as follows,

$$
\sum_{h=h'+1}^{H'}\sum_{k\in\mathcal{S}_h}\frac{L_h}{|\mathcal{S}_h|}(\mu_{k^*}-\mu_{k_h^*})\overset{(a)}{\leqslant}\sum_{h=h'+1}^{H'}\sum_{k\in\mathcal{S}_h}\frac{L_h}{|\mathcal{S}_h|}\frac{2CK}{L_0 2^{h'+1}}
$$

$$
=\sum_{h=h'+1}^{H'}\frac{2CKL_h}{L_0 2^{h'+1}} \tag{13}
$$

$$
=2CK\sum_{h=h'+1}^{H'}2^{h-(h'+1)}
$$

$$
\overset{(b)}{\leqslant}KC^2,
$$

where inequality (a) is due to (9), and inequality (b) is because $H'-h'\leqslant H-h'\leqslant\log_2 C$.

Hence, substituting (12) and (13) into the regret bound, we have the pseudo-regret bound with a probability of at least $1-\delta$ as follows,

$$
\bar{R}_T\leqslant L_0 + 4g\sqrt{2K\log(2KH/\delta)T} + 4\sqrt{T}\log_2 T + 4KC\log_2 T + KC^2
$$

$$
=O\left(\sqrt{KT\log\left(\frac{K\log T}{\delta}\right)} + \sqrt{T}\log T + KC\log T + KC^2\right).
$$

With a similar analysis as in the proof in Theorem 6, we can show that the realized regret $R_T$ also has the same upper bound (in order) with a probability of at least $1-\delta$. □

# G ADDITIONAL EXPERIMENT DETAILS AND RESULTS

## G.1 ATTACK POLICY ON STOCHASTIC BANDITS

We adopt the attack strategy against UCB-type algorithms in Jun et al. (2018). Initially, the adversary randomly selects one suboptimal arm as the target. At each round $t$, after observing the pulled arm $I_t$ and its reward realization $X_{I_t,t}$, the adversary determines the attack $\alpha_t$ according to Equation (7) in Jun et al. (2018) and returns the attacked reward $\tilde{X}_{I_t,t} = X_{I_t,t} - \alpha_t$ to the learner. We set $\Delta_0 = 0.1$, $\sigma = 1$, and $\delta = 0.05$ in the attack model. The attacker cannot attack anymore if the attack budget $C$ is exhausted.

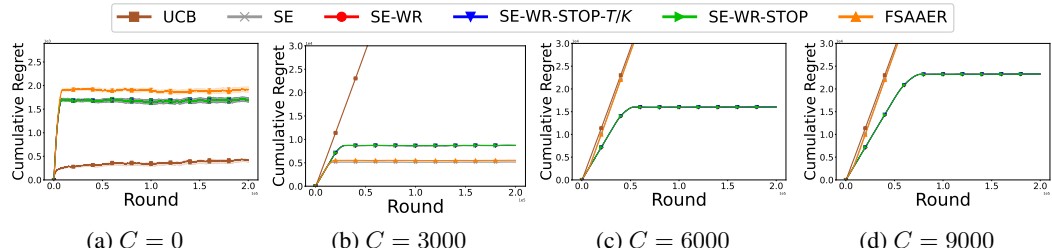

Figure 5: Regret comparison of algorithms with *known* attack budgets when varying budget $C$

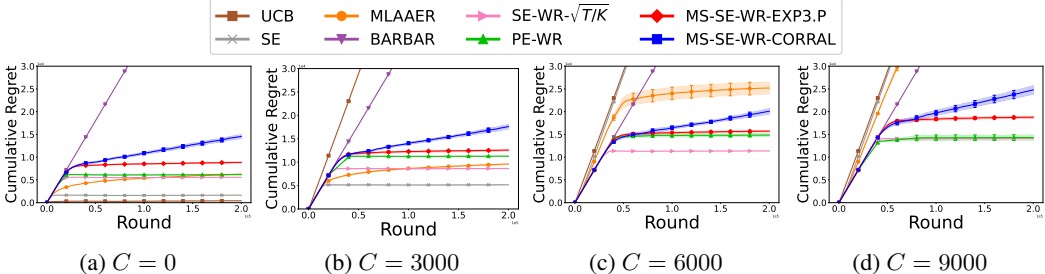

Figure 6: Regret comparison of algorithms with *unknown* attack budgets when varying budget $C$

### G.2 EXPERIMENTS WITH LARGER GAPS

We take $K = 10$ and $T = 200,000$, with rewards for all arms following Bernoulli distributions $\text{Bern}(\mu_i)$ and mean rewards defined as $[0.9, 0.7, 0.65, 0.6, 0.55, 0.5, 0.45, 0.4, 0.35, 0.3]$. We use four attack budgets $C \in \{0, 3000, 6000, 9000\}$. Each experiment is repeated 10 times. Figures 5 and 6 illustrate the cumulative regrets of different algorithms.

With known attack budgets, as depicted in Figure 5, the performance of FSAAER (Lykouris et al., 2018) gradually deteriorates, while our proposed algorithms remain robust as the attack budget $C$ increases. When attack budgets are unknown to the learner, as shown in Figure 6, baselines including MLAAER (Lykouris et al., 2018) achieve sublinear regrets for small budgets (e.g., $C = 0$ and $C = 3000$) as expected, but perform poorly when the budgets become large (e.g., $C = 6000$) and fail entirely when $C = 9000$. As the attack budget $C$ varies, our robust PE-WR and MS-SE-WR algorithms consistently maintain sublinear regrets.

Developing these two STOP algorithms for the known budget case is mainly for (1) later algorithm design for the unknown budget case and (2) the theoretical interest of deriving gap-independent and multiplicative $C$ bounds. The worst performance of the two STOP algorithms when $C = 0$ is expected. Because both STOP conditions are triggered when there are still some suboptimal arms in the candidate arm set. Then, after the STOP conditions, the algorithms uniformly pull these remaining arms, resulting in a linear increasing in regret. Although the regret increases in linear in the later stage of the plot, it is still logarithmic bounded theoretically. Because the remaining suboptimal arms are with reward means close to the optimal arm, which only incurs a slow linear regret increase. On the other hand, note that the y-axis in Figure 3(a) is in $10^3$ scale while the scales of Figures 3(b)(c)(d) are in $10^4$. That is, the two STOP algorithms — although with the worst performance among plots in Figure 3(a) with $C = 0$ — still outperform other scenarios when $C = 3000, 6000, 9000$.

