# OpenReview forum: "Stochastic Bandits Robust to Adversarial Attacks"
_ICLR.cc/2025/Conference — ICLR 2025 Poster_

### Official Review · Reviewer_fffa · 2024-11-04

**Soundness:** 3
**Presentation:** 2
**Contribution:** 2
**Rating:** 6
**Confidence:** 4

**Summary:**

This paper studies stochastic bandit algorithms which are robust to adversarial attacks under a strong adversary that can see the observed arm before attacking.
The paper considers settings with unknown budget cost or known budget cost $C$.
In the known budget case, they provide a gap dependent $((\frac{K}{\delta}) \log T  + KC))$ upper bound that matches the lower bound. They also give gap-independent extensions with upper bounds of $\tilde{O}(\sqrt{KTC})$ or $\tilde{O}(\sqrt{KT} + KC)$ bound.

For the unknown case, they show two stopping criteria-based algorithms, one with an additive dependence in C: $O(\sqrt{KT} + KC^2)$. They show an algorithm that gets $O(T^\alpha)$ regret without corruptions, must have at least $O(T^\alpha + C^\beta)$ regret for $\beta \geq \frac{1}{\alpha}$, thus this upper bound matches this lower bounds in exponents of $C$ (given that it has $\sqrt{T}$ dependence without $C$). Similarly, they give algorithms with multiplicative dependence on $C$ for the regret, that is  $\tilde{O}(\sqrt{KC}T^{\frac{2}{3}})$ or $\tilde{O}(KC\sqrt{T})$.

The paper also provides experimental evidence showing the effectiveness of their algorithms against the attack strategies developed by Jun et al. 2018 comparing it will other corruption-robust MAB algorithms studies in the literature.

**Strengths:**

1. The paper differentiates between attack and corruption models of manipulating multi-armed bandits. It provides insights into the difference between corruption and attacks in terms of the required corruption/attack budget and thus the increased difficulty in preventing attacks compared to corruption.

2. For the successive elimination algorithm SE-WR with increased confidence, also used in Lykouris et al. (2018), the paper shows a tighter regret bound, by better analysis of the concentration results which leads to $O(KC)$ term instead of a gap dependent term in  Lykouris et al. (2018).

3. The authors also give a gap-independent bound for SE-WR and extend the SE-WR algorithm to work in the unknown attack budget settings. They also provide an analysis of the resulting algorithms.

4. The paper also provides experimental evidence showing the effectiveness of their algorithms against the attack strategies developed by Jun et al. 2018 in comparison with multiple MAP defense strategies proposed in the literature.

**Weaknesses:**

1. The results or the discussion do not clarify whether gap-dependent results can be obtained for the unknown horizon setting.


2. Since this paper focuses on making the distinction between attacks and corruption, it seems the main difference is in the inability to use randomization to reduce the scale of the attack, resulting in the need for deterministic algorithms where potentially any arm suffers from all of the corruptions. Thus in the known corruption level case, the results for the setting are directly implied by earlier. (Although this work does a tighter analysis in terms of gaps). A thorough analysis of the lower bounds comparing the setting rather than just comparing the dependence on $K$  could benefit the reader.

3. Experimental results don't have confidence bars, and in the case of no corruptions with known budgets, the STOP algorithms perform worse than other methods. Some discussion on the performance in the absence of corruption is warranted.

Nit:
1. In general, the writing of the paper is very focused on presenting as many results are possible and is very dense in terms of results. The paper could have been formatted better with more discussions around interpreting the results rather than having so many results in the main paper.

 2. There are some typos and inconsistencies in the theorem statements and proofs. For eg in proof of Lemma 14:
  i) the constants are changed from 36 to 64 in $N_k$.
  ii) Line 798, it should be 'triggered' instead of 'trigger'
   iii) Similarly lines 804 to 807 on page 15 in the same proof have $\delta$ with subscripts.

**Questions:**

1. Can you please mention how the lower bounds change or are implied from the corruptions setting to the attack setting in case of unknown horizons?

2. Can you please explain if the gap-dependent results in the unknown corruption case can be obtained for the algorithms under consideration?

3. Can you please explain why the algorithms potentially perform worse in low corruption settings in the experiments?

---

> ### Author Response · Authors · 2024-11-14
> **On Gap-Dependent Results for the Unknown Corruption Budget Case**
>
> > **Q:** 1. The results or the discussion do not clarify whether gap-dependent results can be obtained for the unknown horizon setting.
>
> > **Q:** Can you please explain if the gap-dependent results in the unknown corruption case can be obtained for the algorithms under consideration?
>
> **A:** No, non-trivial gap-dependent results in the unknown corruption case cannot be obtained for the algorithms considered in this paper.
> This is because in bandit literature, the gap-dependent bounds usually appear in $O\left(\sum_k \frac{\log T}{\Delta_k} \right)$-like logarithmic formulas.
> When it comes to polynomial (in $T$) regret bounds (e.g., the $O(\sqrt{KT})$ term of our $\texttt{PE-WR}$ and $\texttt{MS-PE-WR}$ algorithms for the unknown budget case), this gap-dependent logarithmic term would be dominated by the polynomial terms---even when $\Delta_k$ is very small (by the gap-independent regret conversion)---and would not appear in the regret bounds.
>
> However, the current lower bounds (esp., Theorem 3) do not rule out the possibility of devising other algorithms with gap-dependent results in the unknown corruption case. One possible gap-dependent regret bound that future research may consider is $O\left(\exp(C)\cdot\sum_k \frac{\log T}{\Delta_k} \right)$, where the exponential dependence on $C$ is out of the additive and multiplicative scopes studied in our paper.

---

> ### Author Response · Authors · 2024-11-14
> **On Lower Bound Comparison Between Corruption and Attack Cases**
>
> > **Q:** 2. Since this paper focuses on making the distinction between attacks and corruption, it seems the main difference is in the inability to use randomization to reduce the scale of the attack, resulting in the need for deterministic algorithms where potentially any arm suffers from all of the corruptions. Thus in the known corruption level case, the results for the setting are directly implied by earlier. (Although this work does a tighter analysis in terms of gaps). A thorough analysis of the lower bounds comparing the setting rather than just comparing the dependence on $K$ could benefit the reader.
>
> > **Q:** Can you please mention how the lower bounds change or are implied from the corruptions setting to the attack setting in case of unknown horizons?
>
> **A:** We thank the reviewer for this instructive suggestion. We will add the following discussion to the revised manuscript:
>
> "For the unknown corruption/attack budget case, the lower bound for corruption is $\Omega(C + \text{polylog}(T))$, which is achieved by Liu et al. (2021) in the corruption scenario. However, it is not achievable in the attack scenario (Theorem 3). For instance, in terms of $T$, the *polynomial* achievable additive and multiplicative regret bounds (Propositions 4 and 5) for the attack scenario are worse than the *logarithmic* achievable lower bound $\Omega(C + \text{polylog}(T))$ for the corruption scenario. This highlights the difficulty of designing algorithms for the attack case.
>               Additionally, in terms of $C$, the $\Omega(KC)$ lower bound for the attack scenario refutes the possibility of achieving the $\Omega(C + \text{polylog}(T))$ bound."
>
> - Reference: Junyan Liu, Shuai Li, and Dapeng Li. Cooperative stochastic multi-agent multi-armed bandits robust
> to adversarial corruptions. arXiv preprint arXiv:2106.04207, 2021

---

> ### Author Response · Authors · 2024-11-14
> **On Experiment Results**
>
> > **Q:** 3. Experimental results don't have confidence bars, and in the case of no corruptions with known budgets, the STOP algorithms perform worse than other methods. Some discussion on the performance in the absence of corruption is warranted.
>
> > **Q:** Can you please explain why the algorithms potentially perform worse in low corruption settings in the experiments?
>
> **A:** We thank the reviewer for giving us the opportunity to explain the performance of the STOP algorithms when $C=0$.
>               Developing these two STOP algorithms for the known budget case is mainly for (1) later algorithm design for the unknown budget case
>               and
>               (2) the theoretical interest of deriving gap-independent and multiplicative $C$ bounds.
>
> The worst performance of the two STOP algorithms when $C=0$ is expected.
>               Because both STOP conditions are triggered when there are still some suboptimal arms in the candidate arm set.
>               Then, after the STOP conditions, the algorithms uniformly pull these remaining arms, resulting in a linear increase in regret.
>               Although the regret increases linearly in the later stage of the plot, it is still logarithmic and theoretically bounded.
>               Because the remaining suboptimal arms have high reward means close to the optimal arm, which only incurs a slow linear increase. On the other hand, note that the y-axis in Figure 3(a) is in $10^3$ scale while the scales of Figures 3(b)(c)(d) are in $10^4$.
>               That is, the two STOP algorithms --- although with the worst performance among plots in Figure 3(a) with $C=0$ --- still outperform other scenarios when $C=3000, 6000, 9000$.
>
> In this revision, we added this discussion to Appendix F and revised all the figures in the paper with a confidence-bar plot (e.g., see Figures 3 and 4 of Section 6).

---

> ### Author Response · Authors · 2024-11-14
> **On Writing**
>
> > **Q:** In general, the writing of the paper is very focused on presenting as many results are possible and is very dense in terms of results. The paper could have been formatted better with more discussions around interpreting the results rather than having so many results in the main paper.
>
> **A:** We thank the reviewer for suggesting a better exposition. We agree that this paper would benefit from more discussions around interpreting the results. For the final version, we will clarify the results with additional interpretations as given in the previous answers, e.g., [the response to Reviewer C1Fc](https://openreview.net/forum?id=vOFx8HDcvF&noteId=cfHY7THs7T).
>
> > **Q:** There are some typos and inconsistencies in the theorem statements and proofs. For eg in proof of Lemma 14: i) the constants are changed from 36 to 64 in $N_k$. ii) Line 798, it should be 'triggered' instead of 'trigger' iii) Similarly lines 804 to 807 on page 15 in the same proof have $\delta$ with subscripts.
>
> **A:** We thank the reviewer for highlighting these typos and inconsistencies. We corrected the mentioned typos and made multiple careful passes in revising this paper during the rebuttal.
> Specifically for the examples, i) we will consistently use the constant $64$. ii) We corrected the typo in Line 798. iii) We added the subscripts from $\Delta_k$ in Lines 804-807.

---

### Official Review · Reviewer_C1Fc · 2024-11-05

**Soundness:** 3
**Presentation:** 3
**Contribution:** 2
**Rating:** 6
**Confidence:** 2

**Summary:**

The paper studies the design of stochastic bandits algorithms robust to adversarial attacks. In particular, the paper considers an easier setting in which the learner is aware of the attacker budget, and a harder setting in which the learner is not aware of the attacker budget. These results are complemented by lower bounds. Finally, the authors provide an experimental analysis that shows the effectiveness of the proposed approach.

**Strengths:**

The paper advances the state of the art on algorithms robust to adversarial attacks. The paper is well-written and the relationship/improvement relative to previous work is well described.

**Weaknesses:**

The technical contribution is quite weak. For instance, the algorithmic approaches follow previous work and the analysis is not very involved.

**Questions:**

None

---

> ### Author Response · Authors · 2024-11-14
> **On Novelty and Contributions**
>
> We sincerely thank the reviewer for their valuable feedback on our work. We want to take this opportunity to address the concerns raised regarding the novelty and contributions of our research.
>
> ## 1. Novelty and Contribution
>
> We want to emphasize that our work is the first to rigorously study the canonical problem of defending against adversarial attacks in the stochastic multi-armed bandit (MAB) setting. Existing literature has primarily focused on related but distinct problems:
>
> - **Attack Perspective:** Prior research, such as Jun et al. (2018), has explored the problem from the attacker's perspective, designing strategies to mislead bandit algorithms.
> - **Defense in Structured Bandits:** Other studies, including Bogunovic et al. (2021), have investigated defenses within structured bandit settings, such as linear bandits.
> - **Weaker Adversary Models:** Work like Lykouris et al. (2018) addresses defense against corruption but under a weaker adversary model (refer to Lines 68--75 in our manuscript for a detailed comparison).
>
> None of these works have directly addressed the core challenge of defending against adversarial attacks in stochastic MABs, which we believe is a significant gap that our research fills.
>
> ## 2. Comprehensive Study of Robust Algorithms
>
> Our work thoroughly investigates robust algorithms for stochastic MABs under an adversarial attack model. We explore known and unknown attack budget scenarios (or "adaptive budget case" as mentioned by Reviewer fgr9). Specifically, we analyze four distinct cases across Sections 4.1, 4.2, 5.1, and 5.2, covering:
>
> - **Additive Regret Bounds** with respect to the known attack budget $C$.
> - **Multiplicative Regret Bounds** with respect to known $C$.
> - **Additive Regret Bounds** with respect to the unknown attack budget $C$.
> - **Multiplicative Regret Bounds** with respect to unknown $C$.
>
> For each case, we develop algorithms that achieve tight regret upper bounds, matching the corresponding lower bounds detailed in Section 3. The significance of these results is outlined in Table 1 and further discussed in our response to Reviewer MtDK.
>
> ## 3. Algorithmic Innovations and Improvements
>
> While our algorithms draw inspiration from existing techniques, we have made several crucial modifications to adapt them to the adversarial attack model. These adjustments have resulted in tighter bounds than previous work in weaker corruption models or structured bandits. Key improvements include:
>
> - **Removal of $1/\Delta_k^2$ Dependence:** We eliminated the $1/\Delta_k^2$ dependence in the attack term $C$ from the original SE-WR upper bound analysis under the weaker corruption setting Lykouris et al. (2018). This was achieved through a refined concentration inequality analysis.
> - **Improvement of $K$ Factor:** We improved the additive regret upper bound for the unknown $C$ case (Theorem 9) by reducing a $K$ factor compared to the results derived from adapting a known linear bandit algorithm to MAB (Bogunovic et al., 2021). This was made possible by a new algorithm design inspired by multi-phase elimination and tighter concentration analysis.
> - **Enhanced Multiplicative Regret Bounds:** For the unknown $C$ case, we achieved improved multiplicative regret upper bounds of $\tilde{O}(KC\sqrt{T})$ and $\tilde{O}(\sqrt{KC}T^{2/3})$ (Theorem 10), which offer much better $T$ dependence than prior results from Lipschitz bandit algorithms (Kang et al., 2024). This improvement stemmed from our more precise multiplicative regret analysis for the known $C$ case, which involved careful threshold selection in the SE-WR stopping condition.
>
> Additionally, we also present several new lower bounds in the adversarial attack model (Propositions 2, 4, 5), each matching the corresponding improved upper bounds presented in our paper.

---

### Official Review · Reviewer_xPHi · 2024-11-08

**Soundness:** 3
**Presentation:** 3
**Contribution:** 3
**Rating:** 6
**Confidence:** 4

**Summary:**

The paper investigates the classical MAP problem in the adversarial attack setting. The authors provide several tight results covering the case when the attack budget is known/unknown, multiplicative and additive bounds, as well as lower bounds.

**Strengths:**

- This paper addresses a gap in the literature, recognizing that adversarial attacks have not been thoroughly explored within the classical multi-armed bandit (MAB) framework and effectively filling this gap.
- The authors examine both additive and multiplicative bounds, providing a clear comparison that shows which approach performs better based on the attack budget C.
- Figures 1 and, especially, Figure 2 nicely illustrate the results of attack-based multiplicative and additive bounds, offering a well-structured presentation that I haven't seen in comparable works with this level of detail.
- I also like seeing the clear separation between corruption and attack results/settings in one place.
- The paper presents novel findings and situates them within the existing literature, demonstrating that the derived upper bounds are tight (known C case).

**Weaknesses:**

1. Algorithm Design: I didn’t notice any novel or original elements in terms of algorithm design. The PE algorithm has been applied in this context in prior work (cited below), and the idea of using CORRAL has already been explored in similar settings, such as in Misspecified Gaussian Process Bandit Optimization. However, I only find this to be a minor weakness of the paper.

2. Terminology: I like the terminology of “attacks” to distinguish it from the classical “corrupted” setting. However, if the authors intend to introduce this terminology shift, they should properly credit the original paper that first explored this setting and provided robust algorithms: “Corruption-Tolerant Gaussian Process Bandit Optimization.” To my knowledge, this was the first work to present robust algorithms for scenarios in which the attacker can observe the learner's decisions.

3. Literature Review: The literature review in this paper can be improved. The reference section is also too brief and lacks organization. For example, Bogunovic et al. (2020), as cited, do not address the linear setting; this is covered in other relevant papers that are not cited, such as Stochastic Linear Bandits Robust to Adversarial Attacks and A Robust Phased Elimination Algorithm for Corruption-Tolerant Gaussian Process Bandits.

4. Lower Bound Claim: The paper claims that the lower bound result of Ω(KC) is new; however, this result is already established in Stochastic Linear Bandits Robust to Adversarial Attacks (see Appendix C.3). The proof and exposition provided here are quite similar to those in the mentioned paper.

5. Venue Suitability: I’m not entirely sure this paper is a strong fit for ICLR, as I’m not aware of similar works published at this venue previously. This is a consideration for the authors, as they might find broader reach at an alternative venue.

6. Clarity of Comparison (Lines 417-422): The comparison in this section is unclear, and I would appreciate a clearer exposition/steps, especially since the reference provided here is incorrect.

**Questions:**

See 4 and 6 in the Weaknesses section.

---

> ### Author Response · Authors · 2024-11-14
> **On Algorithm Design**
>
> > **Q:** Algorithm Design: I didn’t notice any novel or original elements in terms of algorithm design. The PE algorithm has been applied in this context in prior work (cited below), and the idea of using CORRAL has already been explored in similar settings, such as in Misspecified Gaussian Process Bandit Optimization. However, I only find this to be a minor weakness of the paper.
>
> **A:** We thank the reviewer for this feedback. While our algorithms draw inspiration from existing techniques, we have made several crucial modifications to adapt them to the adversarial attack model. These adjustments have resulted in tighter bounds than previous work in weaker corruption models or structured bandits. Key improvements include:
>
> - **Phase Elimination with Tighter Concentration Radius ($\texttt{PE-WR}$); An improvement of $K$ Factor:** We improved the additive regret upper bound for the unknown $C$ case (Theorem 9) by reducing a $K$ factor compared to the results derived from adapting a known linear bandit algorithm to MAB (Bogunovic et al., 2021). This was made possible by a new algorithm design inspired by multi-phase elimination and tighter concentration analysis.
> - **New Stopping Condition for $\texttt{SE-WR}$ in Model Selection (e.g., CORRAL); Improved Multiplicative Regret Bounds:** For the unknown $C$ case, we achieved improved multiplicative regret upper bounds of $\tilde{O}(KC\sqrt{T})$ and $\tilde{O}(\sqrt{KC}T^{2/3})$ (Theorem 10), which offer much better $T$ dependence than prior results from Lipschitz bandit algorithms (Kang et al., 2024). This improvement stemmed from our more precise multiplicative regret analysis for the known $C$ case, which involved careful threshold selection in the $\texttt{SE-WR}$ stopping condition.
>
> Additionally, we also present several new lower bounds in the adversarial attack model (Propositions 2, 4, 5), each matching the corresponding improved upper bounds presented in our paper, highlighting the optimality of our algorithm design.

---

> ### Author Response · Authors · 2024-11-14
> **On Terminology**
>
> > **Q:** Terminology: I like the terminology of “attacks” to distinguish it from the classical “corrupted” setting. However, if the authors intend to introduce this terminology shift, they should properly credit the original paper that first explored this setting and provided robust algorithms: “Corruption-Tolerant Gaussian Process Bandit Optimization.” To my knowledge, this was the first work to present robust algorithms for scenarios in which the attacker can observe the learner's decisions.
>
> **A:** We thank the reviewer for suggesting a more precise credit to the original robust work. We added the update to the revised manuscript accordingly:
>
> "Bogunovic et al. (2020) are the first to devise robust algorithms for bandits under attacks (referred to as corruptions)."

---

> ### Author Response · Authors · 2024-11-14
> **On Literature Review**
>
> > **Q:** Literature Review: The literature review in this paper can be improved. The reference section is also too brief and lacks organization. For example, Bogunovic et al. (2020), as cited, do not address the linear setting; this is covered in other relevant papers that are not cited, such as Stochastic Linear Bandits Robust to Adversarial Attacks and A Robust Phased Elimination Algorithm for Corruption-Tolerant Gaussian Process Bandits.
>
> **A:** We appreciate the reviewer’s information on this point. We did notice the other two papers
> mentioned by the reviewer during the preparation of our manuscript but missed them due to
> their similar bibliography key strings. We added these references in the revised manuscript
> accordingly as follows,
>
> “Apart from the MAB model, there are works studying robust algorithms on structured ban-
> dits under attacks, e.g., linear bandits (Bogunovic et al., 2021; He et al., 2022), Gaussian
> bandits (Bogunovic et al., 2020; 2022), and Lipchitz bandits (Kang et al., 2024).”

---

> ### Author Response · Authors · 2024-11-14
> **On Lower Bound Claim**
>
> > **Q:** Lower Bound Claim: The paper claims that the lower bound result of $\Omega(KC)$ is new; however, this result is already established in Stochastic Linear Bandits Robust to Adversarial Attacks (see Appendix C.3). The proof and exposition provided here are quite similar to those in the mentioned paper.
>
> **A:** We thank the reviewer for pointing out the similarity between our lower bound and the one in the Stochastic Linear Bandits Robust to Adversarial Attacks paper. After carefully reviewing the two proofs, we agree that the lower bound result of $\Omega(KC)$ is very close to the one in the mentioned paper. We revised the manuscript to clarify this:
>
> "... originally derived in the linear bandit scenario by Bogunovic et al. (2021, Theorem 3)."
>
> We note that as the lower bound (Theorem 1) is not the main contribution of our paper, the novelty of our work remains intact.

---

> ### Author Response · Authors · 2024-11-14
> **On Venue Suitability**
>
> > **Q:** Venue Suitability: I’m not entirely sure this paper is a strong fit for ICLR, as I’m not aware of similar works published at this venue previously. This is a consideration for the authors, as they might find broader reach at an alternative venue.
>
> **A:** We thank the reviewer for this suggestion. Although there are other learning venues that this paper can find broader attention, we believe that ICLR is well-suited to this work as some prior works on attack/robust algorithms for bandits were published in ICLR, e.g.,
> - Ma, Yuzhe, and Zhijin Zhou. ”Adversarial Attacks on Adversarial Bandits.” ICLR, 2023.
> - Buening, Thomas Kleine, et al. ”Bandits Meet Mechanism Design to Combat Click-bait in Online Recommendation.” ICLR, 2024. *(Spotlight Poster)*

---

> ### Author Response · Authors · 2024-11-14
> **On Clarity of Comparison**
>
> > **Q:** Clarity of Comparison (Lines 417-422): The comparison in this section is unclear, and I would appreciate a clearer exposition/steps, especially since the reference provided here is incorrect.
>
> **A:** We thank the reviewer for providing the chance to clarify the comparison in Lines 417-422. In this revision, we further elaborated on the comparison of the three key terms of Theorem 2 of Bogunovic et al. (2021), as follows,
>
> "The most related to ours is the phase elimination algorithm proposed in Bogunovic et al. (2021) for linear bandits. However, directly reducing their results (Bogunovic et al., 2021, Theorem 2)  to our stochastic bandit case would yield a $\tilde{O}(\sqrt{KT} + KC\sqrt{K}\log T+ C^2)$ upper bound for $C\le {\sqrt{T}}/{(K\log(\log K)\log T)}$ only, and, for general $C$, their results would become $\tilde{O}(\sqrt{KT} + KC\sqrt{K}\log T + K^2C^2)$. Our result in Theorem 8, simplified as $\tilde{O}(\sqrt{KT} + KC\log T + KC^2)$, is tighter than theirs by a factor of $\sqrt K$ on the second $\log T$ term, and, for the general $C$ case, our result is tighter by a factor of $K$ on the third $C^2$ term."

---

> > ### Comment · Reviewer_xPHi · 2024-11-27
> > **Re rebuttal**
> >
> > Thank you for the provided answers; I have read them and I will maintain my current positive score.

---

### Official Review · Reviewer_3c3u · 2024-11-22

**Soundness:** 4
**Presentation:** 3
**Contribution:** 3
**Rating:** 8
**Confidence:** 2

**Summary:**

The paper develops stochastic multi-armed bandit algorithms that are robust against adversarial attacks. These attacks can alter the reward the learner observes, and the adversary decides to alter the reward with full knowledge of the action selected by the learner and the realization of the corresponding reward. This way, the model introduces complexities beyond the corruption model, where the adversary has to decide whether or not to corrupt the rewards before having this piece of information. The authors present algorithms for both known and unknown attack budgets with additive and multiplicative regret bounds with respect to the attack budget, providing theoretical proofs of tightness and empirical validations.

**Strengths:**

1) The paper explores an under-studied area of stochastic bandits where adversarial attacks are present and obtains novel results as well as improving prior work about the already studied corruption model.
2) The theoretical bounds are tight (up to log terms), with mathematical proofs for each statement.
3) Experimental results to validate the theoretical claims are provided.
4) The authors do a good job clarifying the differences between corruption and attack models, highlighting the need for specialized approaches.

**Weaknesses:**

1) The implications for practical settings, such as recommendation systems or online auctions, could use some expanding.
2) Unfortunately, all the proofs are relegated to the appendix.

**Questions:**

1) Is the uniqueness of the best arm really needed? What happens if this assumption is not fullfilled?

2) Could you provide concrete applications addressed by your model?

There are a few typos:

091 - To address with MAB with -> To address the MAB with
103 - Show that which type -> Show which type
185 - Lipchitz -> Lipschitz

**Details Of Ethics Concerns:**

The paper is a theoretical one and addresses a generalization of the MAB model.

---

> ### Author Response · Authors · 2024-11-22
> **On Applications**
>
> > **Q:** The implications for practical settings, such as recommendation systems or online auctions, could use some expanding.
>
> > **Q:** Could you provide concrete applications addressed by your model?
>
> **A:** We appreciate the reviewer’s suggestion to expand the discussion on the application.
> We added the following two applications to Appendix A in the revised manuscript. In both scenarios, our robust algorithms provide powerful tools for platforms and network managers to defend against such adversarial attacks effectively.
>
> "A key application of our model lies in online news recommendation systems. Here, the platform (learner) recommends articles (arms) to users with the goal of maximizing total clicks (rewards). However, click fraud botnets (adversaries) may maliciously target the system by generating fake clicks on specific articles (attack reward observations), thereby misleading the recommendation algorithm. This scenario extends naturally to other online recommendation systems, such as e-commerce, video streaming, and social media platforms.
>
> Another potential application is resource allocation for content delivery networks (CDNs). In this context, the CDN manager (learner) allocates resources (e.g., storage, bandwidth) to various servers (arms) to optimize overall performance (rewards, such as response time or throughput). However, the manager might encounter distributed denial-of-service (DDoS) attacks (adversaries) targeting specific servers (attack their reward observations) to degrade system performance."

---

> ### Author Response · Authors · 2024-11-22
> **On Unique Optimal Arm Assumption**
>
> > **Q:** Is the uniqueness of the best arm really needed? What happens if this assumption is not fulfilled?
>
> **A:** We sincerely thank the reviewer for highlighting the unique optimal arm assumption. Upon carefully reviewing our algorithm design and analysis, we confirmed that this assumption is not required for any of our algorithms.
>
> Since our algorithms are based on elimination, they naturally handle cases with multiple optimal arms. Even when multiple arms share the optimal value, the algorithms eliminate suboptimal arms while retaining all optimal arms in the candidate set. The presence of multiple optimal arms in the candidate set does not lead to any additional regret.
>
> For simplicity of presentation, we have retained the current assumption in the paper but have added a footnote in the revised version to clarify this point.

---

> ### Author Response · Authors · 2024-11-22
> **On Writing**
>
> > **Q:** Unfortunately, all the proofs are relegated to the appendix.
>
> **A:** Due to space constraints in the main paper, we have included the proofs in the appendix. For a future arXiv or journal version, we plan to incorporate the proofs directly into the main text.
>
> > **Q:** There are a few typos: 091 - To address with MAB with -> To address the MAB with 103 - Show that which type -> Show which type 185 - Lipchitz -> Lipschitz
>
> **A:** We thank the reviewer for pointing out these typos. We have corrected the identified errors and will conduct thorough reviews during the remaining rebuttal period and after the rebuttal to ensure accuracy.

---

### Meta-Review · Area_Chair_JayL · 2024-12-11

**Metareview:**

There is a general consensus (although mostly not especially strong) that this work provides sufficiently strong results on robust bandits, particularly distinguishing whether or not the attacker knows the exact action taken.  The authors may want to consider mentioning the concurrent work “Corruption-Robust Linear Bandits: Minimax Optimality and Gap-Dependent Misspecification” which also focuses a lot on this distinction, albeit in the slightly different context of linear bandits.

**Additional Comments On Reviewer Discussion:**

The amount of discussion was quite low, but at least no significant concerns seemed to remain afterwards.

---

### Decision · Program_Chairs · 2025-01-22

Accept (Poster)